# Systematic functional analysis of rab GTPases reveals limits of neuronal robustness to environmental challenges in flies

Friederike E Kohrs[1†], Ilsa-Maria Daumann[1†], Bojana Pavlovic[2],
Eugene Jennifer Jin[1‡], F Ridvan Kiral[1§], Shih-Ching Lin[3], Filip Port[2],
Heike Wolfenberg[1], Thomas F Mathejczyk[1], Gerit A Linneweber[1],
Chih-Chiang Chan[3], Michael Boutros[2], P Robin Hiesinger[1]*

[1]Division of Neurobiology, Institute for Biology, Freie Universität Berlin, Berlin, Germany; [2]German Cancer Research Center (DKFZ), Division of Signaling and Functional Genomics and Heidelberg University, Heidelberg, Germany; [3]National Taiwan University, Taipei, Taiwan

**Abstract** Rab GTPases are molecular switches that regulate membrane trafficking in all cells. Neurons have particular demands on membrane trafficking and express numerous Rab GTPases of unknown function. Here, we report the generation and characterization of molecularly defined null mutants for all 26 *rab* genes in *Drosophila*. In flies, all *rab* genes are expressed in the nervous system where at least half exhibit particularly high levels compared to other tissues. Surprisingly, loss of any of these 13 nervous system-enriched Rabs yielded viable and fertile flies without obvious morphological defects. However, all 13 mutants differentially affected development when challenged with different temperatures, or neuronal function when challenged with continuous stimulation. We identified a synaptic maintenance defect following continuous stimulation for six mutants, including an autophagy-independent role of *rab26*. The complete mutant collection generated in this study provides a basis for further comprehensive studies of Rab GTPases during development and function in vivo.

**\*For correspondence:**
prh@zedat.fu-berlin.de

[†]These authors contributed equally to this work

**Present address:** [‡]University of California, San Diego, La Jolla, United States; [§] Yale University, New Haven, United States

## Introduction

Rab GTPases have been named for their initial discovery in brain tissue (*Ra*s-like proteins from rat *brain*), where their abundance and diversity reflect neuronal adaptations and specialized membrane trafficking (*Kiral et al., 2018*; *Touchot et al., 1987*). Yet, Rabs are found in all eukaryotic cells, where they function as key regulators of membrane trafficking between various membrane compartments (*Pfeffer, 2017*; *Zhen and Stenmark, 2015*). Consequently, Rab GTPases are commonly used as markers, and some have become gold standard identifiers of various organelles and vesicles in endocytic and secretory systems (*Pfeffer, 2017*; *Zerial and McBride, 2001*).

Over the years, Rab GTPases have repeatedly been analyzed as a gene family to gain insight into membrane trafficking networks (*Best and Leptin, 2020*; *Chan et al., 2011*; *Dunst et al., 2015*; *Gillingham et al., 2014*; *Gurkan et al., 2005*; *Harris and Littleton, 2011*; *Jin et al., 2012*; *Pfeffer, 1994*; *Stenmark, 2009*; *Zerial and McBride, 2001*). Nonetheless, a complete and comparative null mutant analysis of all family members is currently not available for any multicellular organism. The *Drosophila* genome contains 31 potential *rab* or *rab*-related genes, of which 26 have been confirmed to encode protein-coding genes (*Chan et al., 2011*; *Dunst et al., 2015*; *Jin et al., 2012*), compared to 66 *rab* genes in humans (*Gillingham et al., 2014*) and 11 Rab-related *ypt* genes in

yeast (*Grosshans et al., 2006*; *Pfeffer, 2013*). Of the 26 *Drosophila rab* genes, 23 have direct orthologs in humans that are at least 50% identical at the protein level, indicating high evolutionary conservation (*Chan et al., 2011*; *Zhang et al., 2007*).

In the nervous system, Rab GTPases have been predominantly associated with functional maintenance and neurodegeneration (*Kiral et al., 2018*; *Veleri et al., 2018*). For example, mutations in *rab7* cause the neuropathy CMT2B (*Cherry et al., 2013*; *Spinosa et al., 2008*; *Verhoeven et al., 2003*), Rab10 and other Rabs are phosphorylation targets of the Parkinson's Disease-associated kinase LRRK2 (*Dhekne et al., 2018*; *Steger et al., 2017*), and Rab26 and Rab35 have been implicated in synaptic vesicle recycling (*Binotti et al., 2015*; *Sheehan et al., 2016*; *Uytterhoeven et al., 2011*). Neuronal longevity and morphological complexity have been suggested to require specific Rab-mediated membrane trafficking (*Jin et al., 2018a*; *Jin et al., 2018b*).

We have previously developed a transgenic *Drosophila rab*-Gal4 collection based on large genomic fragments and a design for subsequent homologous recombination to generate molecularly defined null mutants (*Chan et al., 2011*; *Jin et al., 2012*). Analyses of the cellular expression patterns and subcellular localization based on YFP-Rab expression under endogenous regulatory elements by us and others (*Dunst et al., 2015*) have revealed numerous neuronal Rabs with synaptic localization (*Chan et al., 2011*). We originally found that all 26 *Drosophila* Rab GTPases are expressed somewhere in the nervous system and half of all Rabs are enriched or strongly enriched in neurons (*Chan et al., 2011*; *Jin et al., 2012*). A more recent collection of endogenous knock-ins identified more varied expression patterns when more tissues were analyzed, but also validated the widespread neuronal and synaptic expression (*Dunst et al., 2015*). The function of most Rabs with high expression in the nervous system is still unknown.

Here, we provide the first comparative null mutant analysis of all *rab* genes in a multicellular organism. We find that viability, development, and neuronal function are highly dependent on environmental conditions in these mutants. Under laboratory conditions, with minimal selection pressure, seven mutants are lethal, one semi-lethal with few male escapers, two are infertile and six are unhealthy based on progeny counts. Remarkably, all 13 nervous system-enriched *rabs* are viable under laboratory conditions. However, all 13 exhibit distinct developmental or functional defects depending on environmental challenges. Our survey of the complete mutant fly collection provides a basis to systematically elucidate Rab-dependent membrane trafficking underlying development and function of all tissues in a multicellular organism.

## Results

### Generation of the *rab* GTPase null mutant collection

Our earlier observation of a synaptic localization of all nervous system-enriched Rabs led us to speculate that many Rab GTPases may serve roles related to neuron-specific development or function (*Chan et al., 2011*). To test this idea, we set out to generate a complete null mutant collection. We have previously published molecularly defined null mutants of *rab27* (*Chan et al., 2011*) and *rab7* (*Cherry et al., 2013*) as Gal4 knock-ins using a BAC recombineering/homologous recombination approach (*Chan et al., 2011*). Seven additional molecularly defined null mutants have previously been reported: *rab1* (*Thibault et al., 2004*), *rab3* (*Graf et al., 2009*), *rab5* (*Wucherpfennig et al., 2003*), *rab6* (*Purcell and Artavanis-Tsakonas, 1999*), *rab8* (*Giagtzoglou et al., 2012*), *rab11* (*Bellen et al., 2004*), and *rab32* (*Ma et al., 2004*). For the remaining 17 *rab* genes, we generated six null mutants as Gal4 knock-ins that replace the endogenous open-reading frames, or the ATG start codon, using homologous recombination; these include *rab2*, *rab4*, *rab19*, *rab30*, *rabX1*, and *rabX6* (*Figure 1A*; *Figure 1—figure supplement 1A–B*). The remaining 11 null mutants were generated using CRISPR/Cas9, including *rab9*, *rab10*, *rab14*, *rab18*, *rab21*, *rab23*, *rab26*, *rab35*, *rab39*, *rab40*, and *rabX4* (*Figure 1A*; *Figure 1—figure supplement 1C–D*). All mutants were molecularly validated as described in the Materials and methods section.

### All nervous system-enriched *rab* mutants are viable under laboratory conditions

All mutant chromosomes were tested for adult lethality in homozygosity. Of the 26 null mutants, seven are homozygous lethal (*rab 1, 2, 5, 6, 7, 8, 11*) and one, *rab35*, is homozygous semi-lethal with

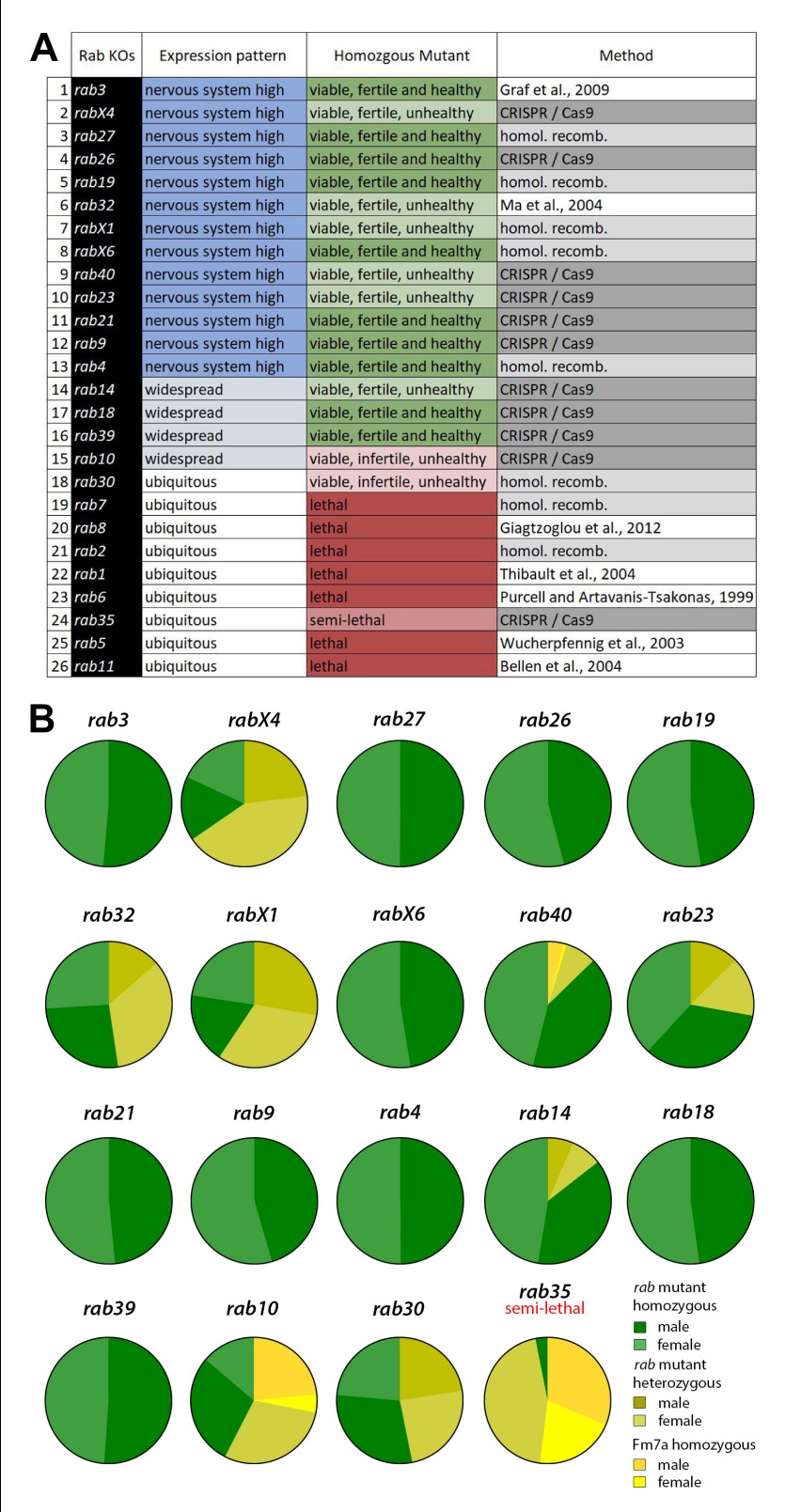

**Figure 1.** Generation and viability analysis of the *rab* null mutant collection. (**A**) List of all 26 *Drosophila rab* null mutants, sorted by expression pattern from 'nervous system-enriched' to ubiquitous based on *Chan et al., 2011*; *Jin et al., 2012*. Two-thirds of the *rab* mutants are homozygous viable and fertile. Eight *rab* mutants are lethal in homozygosity. The origin of the mutants is indicated in the third column. (**B**) Pie charts showing the ratios of

*Figure 1 continued on next page*

*Figure 1 continued*

homozygous versus balanced flies after ten generations. Ten of the 18 viable or semi-lethal *rab* mutants are fully homozygous, while the others still retain their balancer chromosome (shades of yellow) to varying degrees. At least 1000 flies per *rab* mutant were counted.

The online version of this article includes the following figure supplement(s) for figure 1:

**Figure supplement 1.** Design of newly generated *rab* mutants.

**Figure supplement 2.** Pupal expression patterns of nervous system-enriched Rabs based on endogenously tagged Rabs generated by *Dunst et al., 2015*.

**Figure supplement 3.** Adult expression patterns of nervous system-enriched Rabs based on endogenously tagged Rabs generated by *Dunst et al., 2015*.

few male escapers; 18 of the *rab* null mutants are viable as homozygous adults under laboratory conditions (*Figure 1A*).

All mutants were initially generated with the null mutant chromosome in heterozygosity over a balancer chromosome. Balancers contain multiple genetic aberrations, rendering them generally less healthy than wild type chromosomes; balancer chromosomes are therefore outcompeted in healthy stocks after a few generations. However, after 10 generations, only 10 of the 18 viable lines lost the balancer, indicating that eight *rab* mutant chromosomes confer a competitive disadvantage (*Figure 1B*). For five *rab* mutant chromosomes (*rab14, rab23, rab30, rab32*, and *rab40*) a minority of balanced flies remained in the viable stocks, suggesting that the mutant chromosomes in homozygosity are associated with only mildly reduced viability. By contrast, for *rab10, rabX1*, and *rabX4* we found balanced mutant flies in the majority, indicating substantially disadvantageous mutant chromosomes (*Figure 1B*). Sibling crosses between unbalanced homozygous mutant flies revealed an inability to lay eggs for *rab10* mutant flies. In addition, *rab30* mutant males are sterile and crosses of homozygous flies only yield non-developing eggs, a phenotype that was rescued by Rab30 overexpression with the *rab30*-Gal4 line (see Materials and methods). In all other cases, homozygous mutant eggs developed, albeit in some cases at significantly lower numbers or at altered developmental speeds, as discussed in detail below. These observations suggest a range of mutant effects that may affect development or function, yet remain sub-threshold for lethality under laboratory conditions.

Remarkably, all lethal mutants are in *Drosophila rab* genes that are ubiquitously expressed, while all 13 Rab GTPases that we previously reported to be enriched in the nervous system are viable and fertile (*Figure 1A*). This surprisingly binary categorization once again puts a spotlight on the question of specialized Rab GTPase functions in the nervous system. The development and maintenance of the nervous system require robustness to variable and challenging conditions. Endogenous expression patterns based on available knock-ins (*Dunst et al., 2015*) revealed that all 13 nervous-system Rabs are expressed in different patterns in the developing brain (*Figure 1—figure supplement 2*, *Supplementary file 1A*) and in the adult brain (*Figure 1—figure supplement 3*, *Supplementary file 1B*). A comparison of Rab expression in flies with mammalian systems based on published data revealed a high degree of conservation across species, as detailed for each Rab in *Supplementary file 2*. In addition, a comprehensive comparison of functional analyses across species revealed both similarities but also species-specific features of individual Rabs with respect to viability and subcellular localization; this information is presented in detail for each Rab in *Supplementary file 3*. Based on our fly data and these comparisons across species, we hypothesized that many Rabs may provide context-specific neuronal roles that ensure robust development and function that are not apparent under laboratory rearing conditions. To test this hypothesis, we devised a series of assays to test all viable and fertile *Drosophila rab* null mutants for development, function, and maintenance under controlled challenging conditions.

## The majority of viable *rab* mutants affect developmental timing and robustness to different temperatures

First, we analyzed developmental robustness to temperatures at 18°C, 25°C, and 29°C (*Figure 2A–C*). We collected embryos after a 24 hours egg-laying period and measured hatching times of the first 1st instar larvae (*Figure 2D–F*), the first larvae transitioning to pupae (*Figure 2G–I*), and the first adults to eclose (*Figure 2J–L*) at all three temperatures. The 16 homozygous viable and fertile

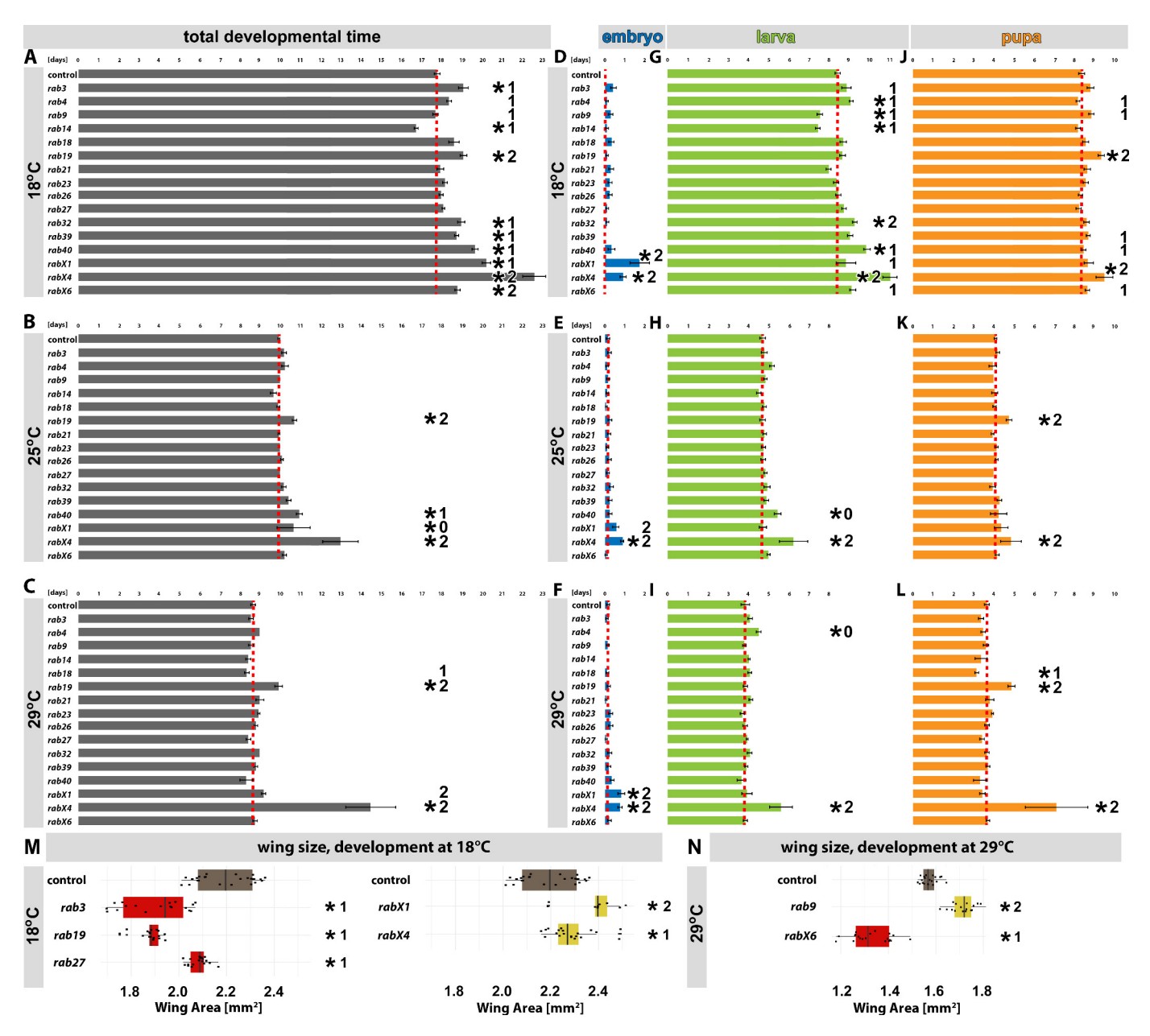

**Figure 2.** Developmental analyses of all viable *rab* mutants at different temperatures. (A–C) Developmental time from embryogenesis to adults at 18°C (A), 25°C (B), and 29°C (C) for all homozygous viable *rab* mutants. (D, G, and J) Developmental time at 18°C for all homozygous viable *rab* mutants, separated into embryonal (blue, D), larval (green, G) and pupal (orange, J) phases. (E, H, and K) Developmental time at 25°C for all homozygous viable *rab* mutants, separated into embryonal (blue, E), larval (green, H) and pupal (orange, K) phases. (F, I, and L) Developmental time at 29°C for all homozygous viable *rab* mutants, separated into embryonal (blue, F), larval (green, I) and pupal (orange, L) phases. (A–L) Dashed red line = mean of control. Mean ± SEM; *p<0.05 (for the specific statistical values see *Figure 2—figure supplement 1*); 0, 1, or 2 indicate if the specific phenotype could not be validated (0), could be validated by either backcrossing or mutant over deficiency (1) or could be validated by both (2); Unpaired non-parametric Kolmogorov-Smirnov test. (M–N) Wing surface area measurement for validated homozygous viable *rab* mutants at 18°C (M) and 29°C (N). Wild type (brown) and *rab* mutant with significantly reduced (red) and increased wing sizes (yellow) compared to control. Boxplot with horizontal line representing the median; individual data points are represented as dots. Fifteen to 22 wings per genotype were quantified; *p<0.05 (for the specific statistical values see *Figure 2—figure supplement 2*); 0, 1, or 2 indicate if the specific phenotype could not be validated (0), could be validated by either backcrossing or mutant over deficiency (1) or could be validated by both (2); ordinary one-way ANOVA with pair-wise comparison.

The online version of this article includes the following figure supplement(s) for figure 2:

**Figure supplement 1.** Validation of developmental timing phenotypes of viable *rab* mutants at different temperatures.

**Figure supplement 2.** Wing surface area measurement for all homozygous viable *rab* mutants at 18°C and 29°C.

*Figure 2 continued on next page*

**Figure supplement 3.** Examples of wing defects after development at different temperatures.

mutants include all 13 nervous system-enriched *rabs* plus *rab14*, *rab18*, and *rab39*. To control for genetic background effects, we further tested all mutants with developmental phenotypes in two additional genetic backgrounds: first, the mutant chromosome in heterozygosity over a genomic deficiency uncovering the respective mutation; second, we backcrossed the mutants for three generations to control flies, thereby making the genetic background >80% identical to the control stock (see Materials and methods). We only considered phenotypes that were validated in at least one of the two additional genetic backgrounds; the number of validations are indicated as a number next to the asterisks marking significant differences in *Figure 2* as well as in detail in *Figure 2—figure supplement 1* and in *Supplementary file 4*.

Of the 16 homozygous viable and fertile mutants, 12 exhibited specific defects in developmental timing and an additional two mutants exhibited defects in wing development as described below. No developmental defects were observed only for *rab21* and *rab26*. The 12 mutants with developmental timing phenotypes exhibited the following phenotypes (in order of severity): *rabX4* exhibited the longest overall developmental delay, including delays of embryo, larval and pupal stages at all three developmental temperatures. *rabX4* mutant flies exhibited normal egg-laying behavior, but most eggs did not develop; only few *rabX4* adult escapers developed with 2–4 days developmental delay (*Figure 2A–L*; *Figure 2—figure supplement 1A–C*; *Supplementary file 4*). *rabX1* was the only mutant that exhibited selective delays of embryo development at all three temperatures, but normal timing of larval and pupal development (*Figure 2D–L*). *rabX1* mutant flies laid very few eggs, with only a subset of these developing to adulthood (20% of control; *Supplementary file 4*). *rab19* was the only mutant that exhibited selective delays of pupal development (but normal embryo and larval development) at all temperatures (*Figure 2D–L*). In addition, *rab19* exhibited a 50–80% rate of late pupal lethality specifically at 29°C, that was not observed at lower temperatures. All *rab19* adults raised at 29°C died within a few days. *rab32* exhibited increased late pupal lethality specifically at 29°C, while survivors exhibited normal eclosion timing. At 18°C, *rab32* mutants exhibited a mild overall developmental delay due to delayed larval development (*Figure 2A,G*). *rab40* exhibited a developmental delay at 18°C (*Figure 2A,G*) that was validated in both alternate genetic backgrounds (*Figure 2—figure supplement 1D,G*), a mild developmental delay at 25°C (*Figure 2B,H*) that was validated in a backcrossed background (*Figure 2—figure supplement 1E*), and no developmental delay at 29°C (*Figure 2C*). *rabX6* exhibited a mild developmental delay only at 18°C (*Figure 2A*) which could be validated in both alternate genetic backgrounds (*Figure 2—figure supplement 1D*). Similarly, *rab39* and *rab3* both exhibited a mild overall developmental delay at 18°C (*Figure 2A*) that were both validated in a backcrossed background (*Figure 2—figure supplement 1D*). *rab4* exhibited mildly delayed overall development at 18°C (*Figure 2A,G*) that was validated in a backcrossed background (*Figure 2—figure supplement 1D,G*). *rab9* and *rab14* were the only mutants with a shorter larval development at 18°C (*Figure 2G*) that was validated over deficiencies in both cases (*Figure 2—figure supplement 1G*). Finally, *rab18* was the only mutant that exhibited shortened pupal development specifically at 29°C (*Figure 2L*) that was validated in a backcrossed background (*Figure 2—figure supplement 1I*).

Taken together, these 12 mutants uncover developmental sensitivities of different developmental stages and with varying temperature-dependencies. Development at 18°C revealed increased variability of developmental timing in the majority of mutants that resulted from variability of larval development which in turn depends on larval behavior (*Figure 2A,G*). In contrast to larval development, pupae did not exhibit an increased variability of developmental timing. Developmental timing at higher temperatures was significantly less variable for all developmental stages. While some prominent developmental delays occurred at all temperatures (*rabX4* and *rabX1*), other mutants were selectively sensitive to development at higher temperatures (*rab18, rab19, rab32*) or lower temperatures (*rab4, rab40*).

Temperature is known to affect organ development through changes in cell size (*Azevedo et al., 2002*). For example, the *Drosophila* wing in control flies is 25–45% larger after development at 18°C compared to development at 29°C (*Figure 2M,N*). As with developmental timing, specific *rab*

mutants exhibited opposite developmental defects either only at lower or higher developmental temperatures. At 18°C we observed significantly smaller wings for *rab3*, *rab19*, and *rab27* and significantly larger wings for *rabX1 and rabX4*, the two mutants with the longest developmental delay at 18°C (*Figure 2A,M*). At 29°C, we found significantly smaller wings in the *rabX6* mutant and significantly larger wings in the *rab9* mutant compared to controls at the same developmental temperature (*Figure 2N*). We only scored phenotypes that were validated in at least one additional genetic background (backcrossed or over deficiency, *Figure 2—figure supplement 2*). Finally, the *rab23* null mutant exhibited a planar cell polarity phenotype of wing bristles reported previously (*Dunst et al., 2015*; *Pataki et al., 2010*). In addition, we observed a previously not reported highly penetrant transversal p-cv vein shortening (in 90% of the wings studied) at 18°C, which was ameliorated at 29°C (12% penetrance) in the *rab23* mutant (*Figure 2—figure supplement 3*). In summary, 14 of the 16 viable and fertile null mutants exhibit specific developmental defects, most of which only occurred (or were significantly exacerbated) at high (29°C) or low (18°C) developmental temperatures.

## A subset of *rab* mutants affect the maintenance of stimulus-dependent synaptic function

To challenge neuronal function and maintenance, we tested the effect of continuous light stimulation on photoreceptor neurons, a widely used model to identify mutants affecting neuronal maintenance and degeneration in *Drosophila* (*Jaiswal et al., 2012*). Electroretinograms (ERGs) are extracellular recordings that reveal two aspects of photoreceptor function: first, the depolarization measures the ability of photoreceptor neurons to convert a light stimulus into an electrical signal; reduced depolarization can be the result of a reduced ability to perceive light (phototransduction), reduced electrical properties of individual cells, or loss of neurons. Second, the ERG 'on' transient indicates the ability to transmit the presynaptic signal to the postsynaptic interneurons. Loss of the 'on' transient can result from defective neurotransmission or degeneration that starts at the synapse, as shown for the *rab7* mutant previously (*Cherry et al., 2013*). The ERG is mostly used as a qualitative method, because both depolarization and 'on' transient intensities are highly sensitive to differences in genetic background, eye pigmentation, intensity of the light stimulus and other recording variables. To identify a sensitive period during which mild alterations of neuronal function and maintenance should be measurable, we established sensitization curves over several days of stimulation. In control flies, continuous stimulation leads to a gradual decline of the 'on' transient amplitude (*Figure 3A*) and depolarization (*Figure 3B*) over a 7-day period. Two days light stimulation represent a highly sensitized period with a dynamic range for improvement or worsening of potential defects for both the 'on' transient (*Figure 3A*) and depolarization (*Figure 3B*).

For all 16 viable and fertile *rabs* plus the two infertile mutants *rab10* and *rab30*, we tested mutants in a *white minus* background (white-eyed flies). First, we performed ERG recordings of newly hatched flies to assess neuronal function immediately after development ('0 day'; *Figure 3C–D*). None of the mutants exhibited significant reductions of their 'on' transient (*Figure 3C*) or depolarization (*Figure 3D*) immediately after hatching (0 day). Next, we used continuous light stimulation to measure changes in function after continuous stimulation (*Figure 3E–F*) and dark-rearing to assess aging in the absence of stimulation (*Figure 3G–H*). After 2 days of light stimulation, six *rab* mutants exhibited significantly reduced neurotransmission compared to control based on their 'on' transients: *rab3*, *rab14*, *rab19*, *rab26*, *rab30* and *rabX6*. For five of these six, the defect was specific to synaptic function without significant effects on depolarization (*rab3*, *rab19*, *rab26*, *rab30*, and *rabX6*, all with nervous system-enriched expression). By contrast, one mutant (*rab14*, with widespread expression) additionally exhibited a significantly decreased depolarization, indicating more generally reduced cellular function. Hence, neuron-enriched expression and synaptic localization of several Rab GTPases correlate with robustness of synaptic function under continuous stimulation.

To test whether these maintenance defects were strictly stimulus-dependent, we tested dark-reared flies. None of the five *rabs* with specific synaptic defects (*rab3*, *rab19*, *rab26*, *rab30* and *rabX6*) exhibited reduced neurotransmission in the absence of a light stimulus. By contrast, *rab14* and additionally *rab27*, exhibited both reduced transmission and depolarization after 4 days in the dark, suggesting stimulus-independent and aging-related defects. These findings indicate that the synaptic defects of *rab3*, *rab19*, *rab26*, *rab30*, and *rabX6* are stimulus-dependent, and the defects of *rab14* and *rab27* aging-dependent functional maintenance defects. A role for *rab27* in neuronal aging has recently been reported (*Lien et al., 2020*).

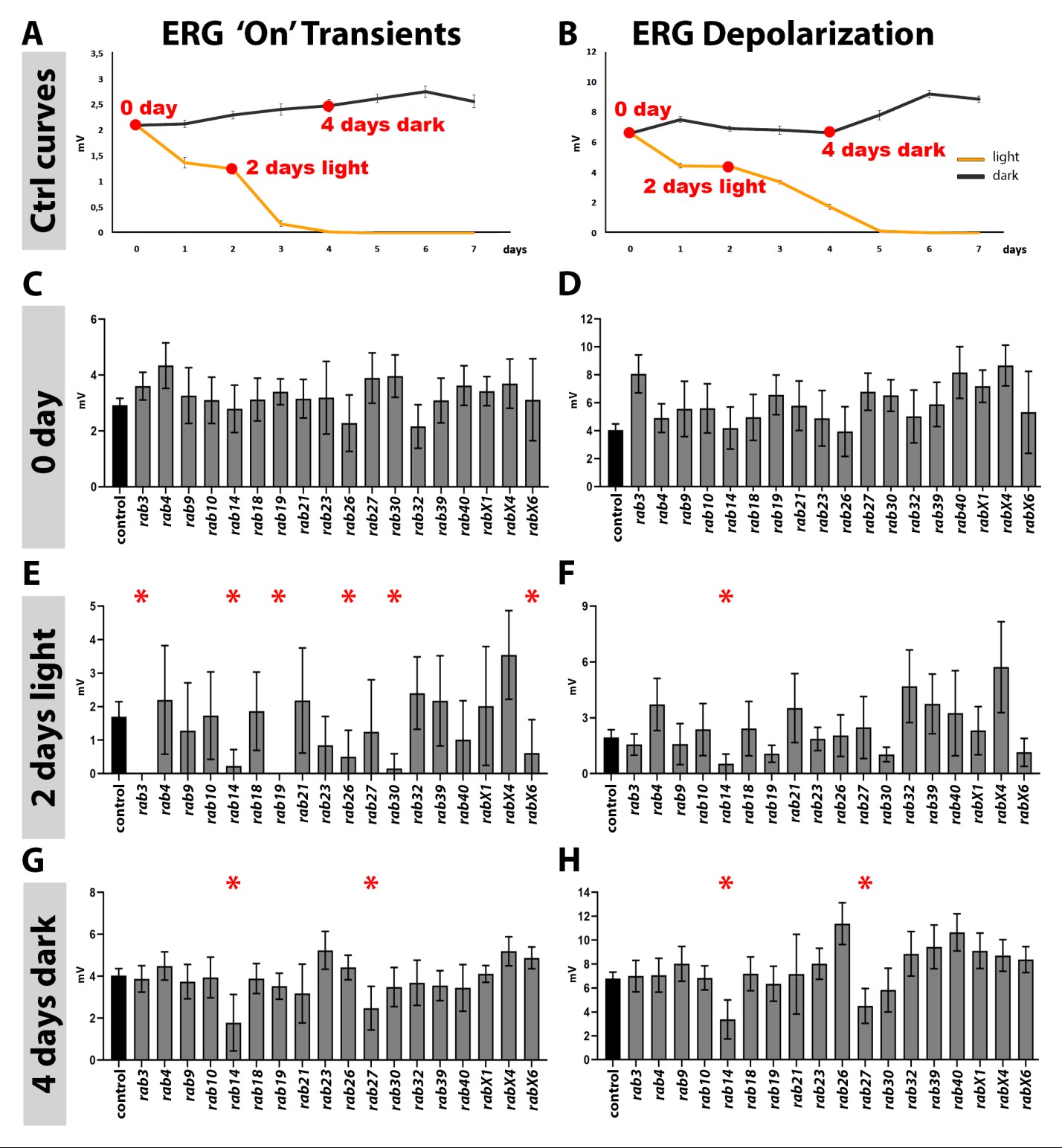

**Figure 3.** Analysis of neuronal function and maintenance based on electroretinograms. (A–B) Sensitization curves for light stimulated (orange curve) and dark-reared (black curve) wild type flies generated by electroretinogram (ERG) recordings. 'on' transient signal is lost after 4 days of light stimulation. Complete loss of depolarization signal after 5 days of light stimulation. 0 day, 2 days light stimulation and 4 days dark-rearing are highlighted in red. Mean ± SEM; 25–30 flies were recorded for each day (0–7 days) and each condition (light and dark); Ordinary one-way ANOVA with pair-wise comparison. (C–D) 'on' transient and depolarization of newly hatched (0 day) flies. Wild type control in black, all homozygous viable *rab* mutants in grey. (E–F) 'on' transient and depolarization of wild type (black) and homozygous viable *rab* mutants (grey) after 2 days of light stimulation. (G–H) 'on'
*Figure 3 continued on next page*

*Figure 3 continued*

transient and depolarization of wild type (black) and homozygous viable *rab* mutants (grey) after 4 days of dark-rearing. (**C–H**) Mean ± SD; *p<0.05; 25–30 flies were recorded for each genotype and condition; ordinary one-way ANOVA with group-wise comparison.

## A subset of *rab* mutants affect in a stimulus-dependent manner the maintenance of rhabdomeres, a high-turnover membrane compartment harboring the phototransduction machinery

During the sensitive period after 2 days of light stimulation, both 'on' transients (*Figure 3E*) and depolarization (*Figure 3F*) exhibited higher variability amongst individuals than before stimulation (*Figure 3C,D*) or after 4 days in the dark (*Figure 3G,H*). This variability after 2 days of light stimulation could be a consequence either of functional differences amongst individuals or of progressive cell death, which is known to be induced by prolonged stimulation of photoreceptor neurons (*Kiselev et al., 2000*; *Xiong and Bellen, 2013*). We tested for programmed cell death using cleaved *Drosophila* death caspase-1 (DCP-1) as an apoptotic marker. None of the 18 viable *rab* mutants exhibited elevated levels of DCP-1 before or after 2 days of light stimulation (*Figure 4A–B*; *Figure 4—figure supplement 1*). As a positive control, we used DCP-1 to visualize retinal degeneration in the *rdgC^{306}* mutant (*Steele and O'Tousa, 1990*; *Figure 4C*). Hence, increased phenotypic variability during this sensitized period likely reflects individual differences of functional and maintenance defects compared to control. Indeed, the co-labeling of rhabdomeres in these experiments revealed highly variable structural defects in *rab* mutant eyes after 2 days of light stimulation. The rhabdomeres are densely stacked membranes that are characterized by large-scale, light-dependent membrane trafficking of rhodopsin and other phototransduction proteins (*Frechter and Minke, 2006*; *Schopf and Huber, 2017*; *Xiong and Bellen, 2013*). We found no rhabdomere defects in any of the 16 viable plus viable but infertile *rab* mutants before stimulation, consistent with the absence of functional defects after development but prior to a functional challenge (*Figure 4A,E*; *Figure 4—figure supplement 1*). By contrast, after 2 days of light stimulation rhabdomere structures exhibited highly increased variability (*Figure 4B,D,F*). In control, rhabdomere area increased on average ~30% after 2 days of stimulation, while seven *rab* mutants exhibited a significant decrease in area greater than the control variability indicated by its standard deviation (*rab4*, *rab18*, *rab21*, *rab27*, *rab30*, *rab32*, *rab40*; *Figure 4D*). In addition, rhabdomere shapes exhibited similarly increased variability and significant changes in three additional rab mutants (*rab19*, *rab23*, and *rab26*; *Figure 4E–F*). We conclude that at least 10 of the 18 viable *rab* mutants affect membrane turnover in rhabdomeres when challenged with continuous stimulation.

## Synaptic maintenance defects in viable *rab* mutants do not coincide with defective autophagy or Rab11-dependent endosomal recycling

Next, we analyzed the morphology of photoreceptor axon projections after light stimulation compared to newly hatched flies using an antibody against the photoreceptor membrane protein Chaoptin. All 13 nervous system-enriched *rab* mutants exhibited axonal projections that were indistinguishable from control in newly hatched flies (*Figure 5—figure supplement 1*). We found no obvious developmental defects amongst newly hatched flies. All except one mutant looked indistinguishable from control; *rabX1* exhibited normal axonal projections, but unusual accumulations of Chaoptin in non-photoreceptor cell bodies surrounding the neuropils (arrowheads in *Figure 5A*), a phenotype previously observed for endomembrane degradation mutants including *rab7* (*Cherry et al., 2013*) and the v-ATPase *v100* (*Williamson et al., 2010*).

After 2 days of light stimulation, two mutants exhibited alterations of their axon terminal morphology. Mutants for *rab26,* and to a lesser extent *rab19*, exhibited distinct membrane accumulations at the distal tips of R1-R6 photoreceptor axon terminals (arrows in *Figure 5A*). Both *rab19* and *rab26* are amongst the five neuronal *rabs* exhibiting stimulus-dependent specific transmission maintenance defects. We next tested whether these membrane accumulations are associated with defects in autophagosome formation or clearance. In wild type flies, Atg8/LC3-positive autophagosomes were relatively infrequent given the number of axon terminals in the lamina both before and after light stimulation (*Figure 5B*). Notably, none of the five neuronal *rab* mutants with synaptic maintenance defects exhibited significantly altered Atg8 labeling. By contrast, in the *rabX1* mutant,

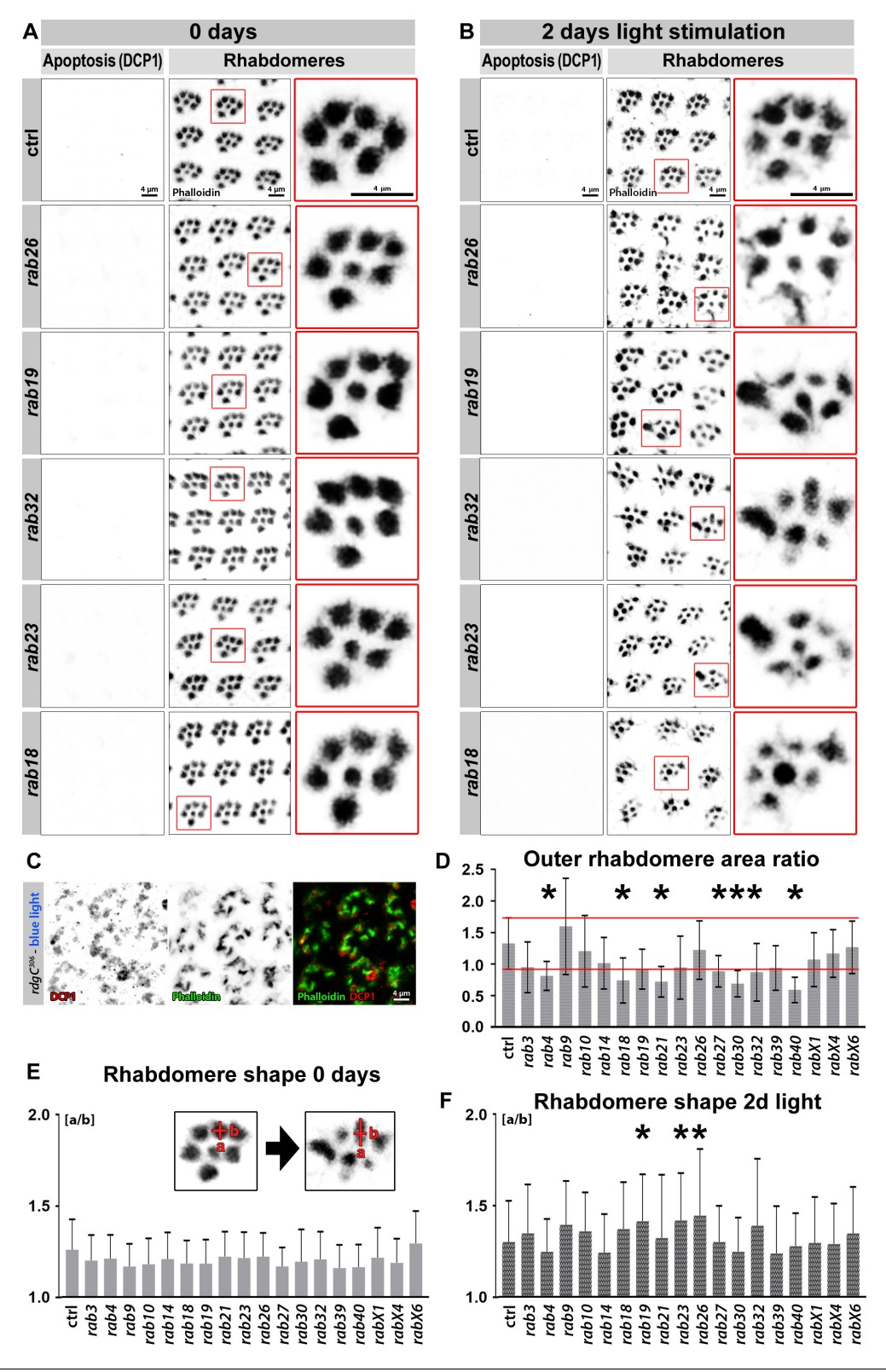

**Figure 4.** Viable *rab* mutants show no apoptosis based on DCP-1 immunolabeling but display morphological changes in rhabdomeres after continuous light stimulation. (A–B) Examples of *rab* mutant retinas which show rhabdomere changes and no increased levels in the apoptotic marker DCP-1 after 2 days of light stimulation compared to control (B) and newly hatched flies (A). Zoom-ins of single ommatidia are highlighted by red boxes. Scale bar = 4 μm; number of retinas n = 5–7 from different animals per antibody staining. (C) *rdgC*[306] mutant ommatidia show high levels of DCP-1 (red) after

*Figure 4 continued on next page*

*Figure 4 continued*

continuous blue light stimulation. Labeling with phalloidin (green) reveals highly disrupted rhabdomere morphology. Scale bar = 4 µm; number of retinas n = 4 per antibody staining. (D) Area ratio of outer rhabdomeres R1-R6. The standard deviation range of wild type control is highlighted by red lines. Outer rhabdomere area ratio was calculated as described in Materials and methods. Mean ± SD; *p<0.05 (only significances outside SD range are marked); number of outer rhabdomeres counted n = 150 from three to six animals. Ordinary one-way ANOVA with group-wise comparison. (E–F) After 2 days of light stimulation outer rhabdomere shape exhibited increased variability (F) compared to newly eclosed flies (E). Outer rhabdomere shape was calculated as described in Materials and methods and examples of single ommatidia (left: 0 day, right: 2 days of light stimulation) are shown in the zoom-ins (E). Mean + SD; *p<0.05; number of outer rhabdomeres counted n = 150 from three to six animals. Ordinary one-way ANOVA with group-wise comparison.

The online version of this article includes the following figure supplement(s) for figure 4:

**Figure supplement 1.** No viable *rab* mutants show apoptosis based on DCP-1 immunolabeling, some display morphological changes in rhabdomeres after 2 days of continuous light stimulation.

Atg8 levels were increased in cell bodies distal of axon terminals already prior to stimulation (arrow-heads in *Figure 5B*). Stimulus-dependent increased numbers of Atg8-positive compartments in axon terminals were observed for *rab23, rab27, rab32,* and as prominent clusters for *rabX1,* none of which exhibited stimulus-dependent synaptic maintenance defects (*Figure 5B*). These observations do not support a link between synaptic maintenance and autophagy based on viable, neuron-enriched Rabs.

We previously showed that most nervous system-enriched Rabs, including Rab19 and Rab26, encode proteins that colocalize with the recycling endosome marker Rab11 at photoreceptor axon terminals (*Chan et al., 2011*). Using the same 2 days light stimulation assay, we found that in wild type, Rab11 is strongly upregulated in the synaptic terminals after stimulation, indicating increased membrane trafficking. Surprisingly, we found the same stimulus-dependent increase of Rab11 as in control in all mutants except *rabX1,* consistent with a recent characterization of RabX1's endolyoso-mal function (*Laiouar et al., 2020*; *Woichansky et al., 2016*, *Figure 5C*). In summary, all Rabs impli-cated in synaptic functional maintenance exhibited Atg8 and Rab11 levels similar to control after light stimulation; our analyses therefore suggest that these Rabs employ mechanisms distinct from canonical Rab11-dependent endomembrane recycling and Atg8-dependent autophagy at synapses.

## Loss of *rab26* does not discernibly affect membrane trafficking associated with synaptic vesicles or autophagy in the adult brain

Rab26 has been proposed to link synaptic vesicle recycling to autophagy based on experiments in mammalian cell culture and *Drosophila* using overexpression of GTP-locked and GDP-locked variants (*Binotti et al., 2015*). Here, we provide an analysis of the *rab26* null mutant. In support of a role of autophagy in synaptic vesicle turnover, we found that *rab26* is one of the *rab* null mutants that exhibit reduced stimulus-dependent functional maintenance (*Figure 3E*), while being one of only two mutants without any developmental defect in our assays (*Figure 2*). In addition, *rab26* null mutant axon terminals exhibited pronounced membrane accumulations after continuous light stimu-lation (*Figure 5A*). However, we found no significant changes of the autophagosomal marker Atg8/LC3 (*Figure 5B*). These findings prompted us to probe putative roles of Rab26 at synaptic terminals in more detail.

Expression of GTP-locked Rab26 in adult photoreceptor neurons led to a complete loss of neuro-transmission, while neither complete loss of *rab26* function nor expression of GDP-locked Rab26 sig-nificantly affected neurotransmission in newly hatched flies (*Figure 6A*). GTP-locked Rab26 protein formed enlarged accumulations as observed in the earlier study. Compartments and accumulations marked by YFP-tagged WT or GTP-locked Rab26 largely exclude synaptic markers (Syt1 and CSP; *Figure 6B–C*) as well as the autophagosome marker Atg8 (*Figure 6D–E*). By contrast, the recycling endosomal markers Rab11 (*Figure 6D–E*) and the endosomal markers Hrs and Syx7 (*Figure 6F–G*) all exhibit elevated levels in axon terminals expressing GTP-locked Rab26. These findings suggest an endosomal role at synaptic terminals that may not be directly linked to synaptic vesicles and autophagy.

Next, we compared the findings from GTP-locked Rab26 overexpression to the *rab26* null mutant. Adult brains mutant for *rab26* did not exhibit obvious changes of Atg8 or Syt1 (*Figure 6H–K*). The null mutant brains appeared morphologically normal and exhibited no difference for any of

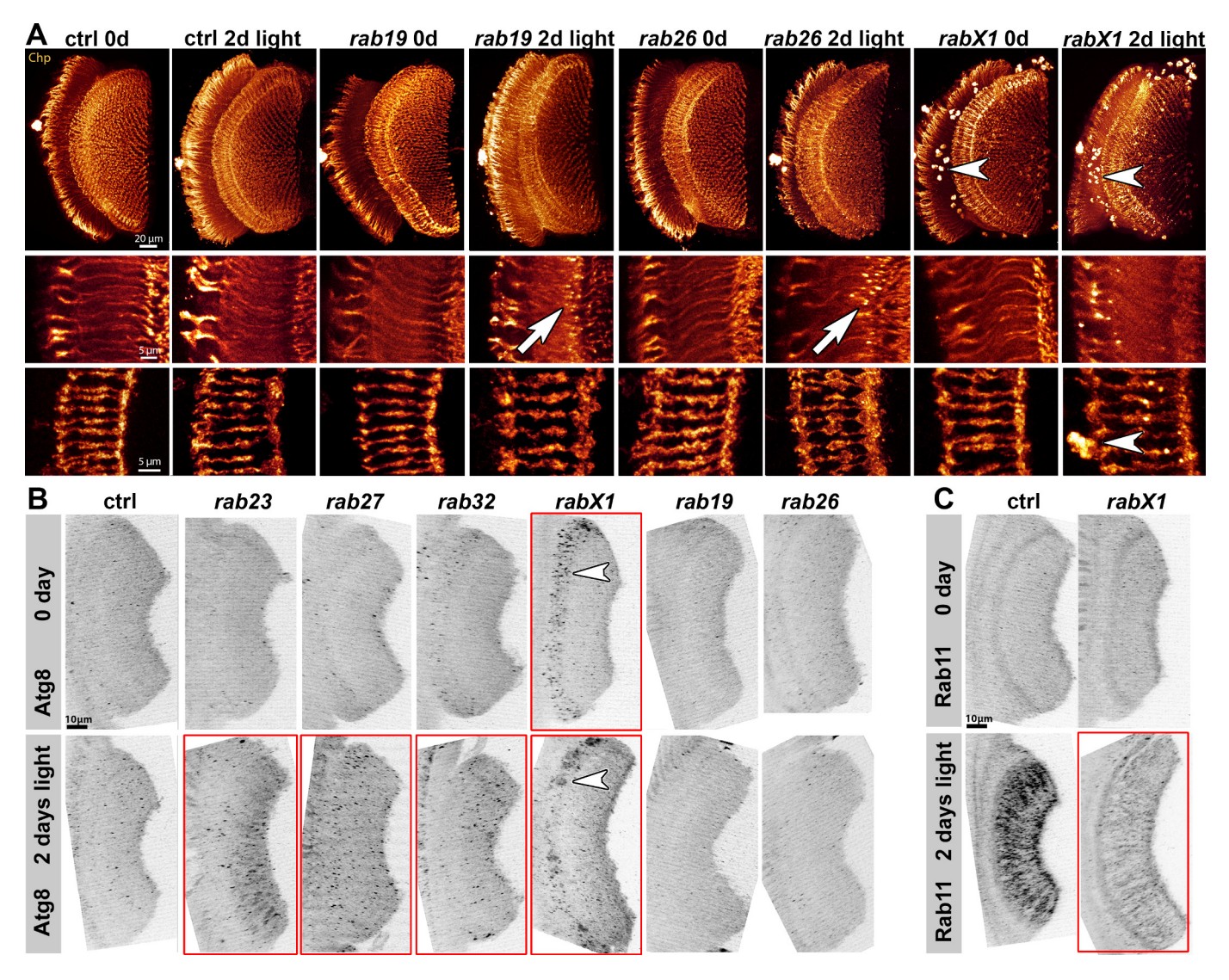

**Figure 5.** Analyses of morphology, recycling endosomal function (Rab11) and autophagy (Atg8) at photoreceptor axon terminals after continuous light stimulation. (**A**) Examples of Chaoptin-labeling (Chp) of 0 day and 2 days light stimulated wild type and *rab* mutant photoreceptor projections (overview top panel, R1-R6 middle panel, R7-R8 bottom panel). The *rabX1* mutant exhibits Chaoptin accumulations in non-photoreceptor cell bodies independent of stimulation (arrowheads). After 2 days of light stimulation, *rab26* and *rab19* mutants display membrane accumulations in their axon terminals (arrows). Scale bar = 20 µm (top panel), 5 µm (middle and bottom panel); number of brains n = 3–5 per antibody staining. (**B**) Examples of Atg8 labeling of photoreceptor projections in retina-lamina preparations of newly hatched and 2 days light stimulated wild type flies and six *rab* mutants. Only *rab23*, *rab27*, and *rab32* show significant increases in Atg8-positive compartments after 2 days of light stimulation (highlighted by red boxes). *rabX1* flies exhibit Atg8-positive compartments in cell bodies (arrowheads). Scale bar = 10 µm; number of retina-lamina preparations n = 3 for each condition and staining. (**C**) Examples of Rab11 labeling of photoreceptor projections in retina-lamina preparations of newly hatched and 2 days light stimulated wild type and *rabX1* flies. Increase in Rab11 levels is suppressed in *rabX1* mutants after 2 days of light stimulation (highlighted by red box). Scale bar = 10 µm; number of retina-lamina preparations n = 3 for each condition and staining.

The online version of this article includes the following figure supplement(s) for figure 5:

**Figure supplement 1.** Systematic analysis of photoreceptor axon morphology of newly eclosed adults and after 2 days of continuous light stimulation.

the markers analyzed above. These findings do not support a strict requirement for any essential endomembrane trafficking process during development and initial function.

*Binotti et al., 2015* focused their *Drosophila* analyses on the larval neuromuscular junction (NMJ), we also investigated *rab26* loss-of-function in presynaptic boutons of these motoneurons and their postsynaptic muscle. We further generated a polyclonal antibody against the cytosolic N-terminus of

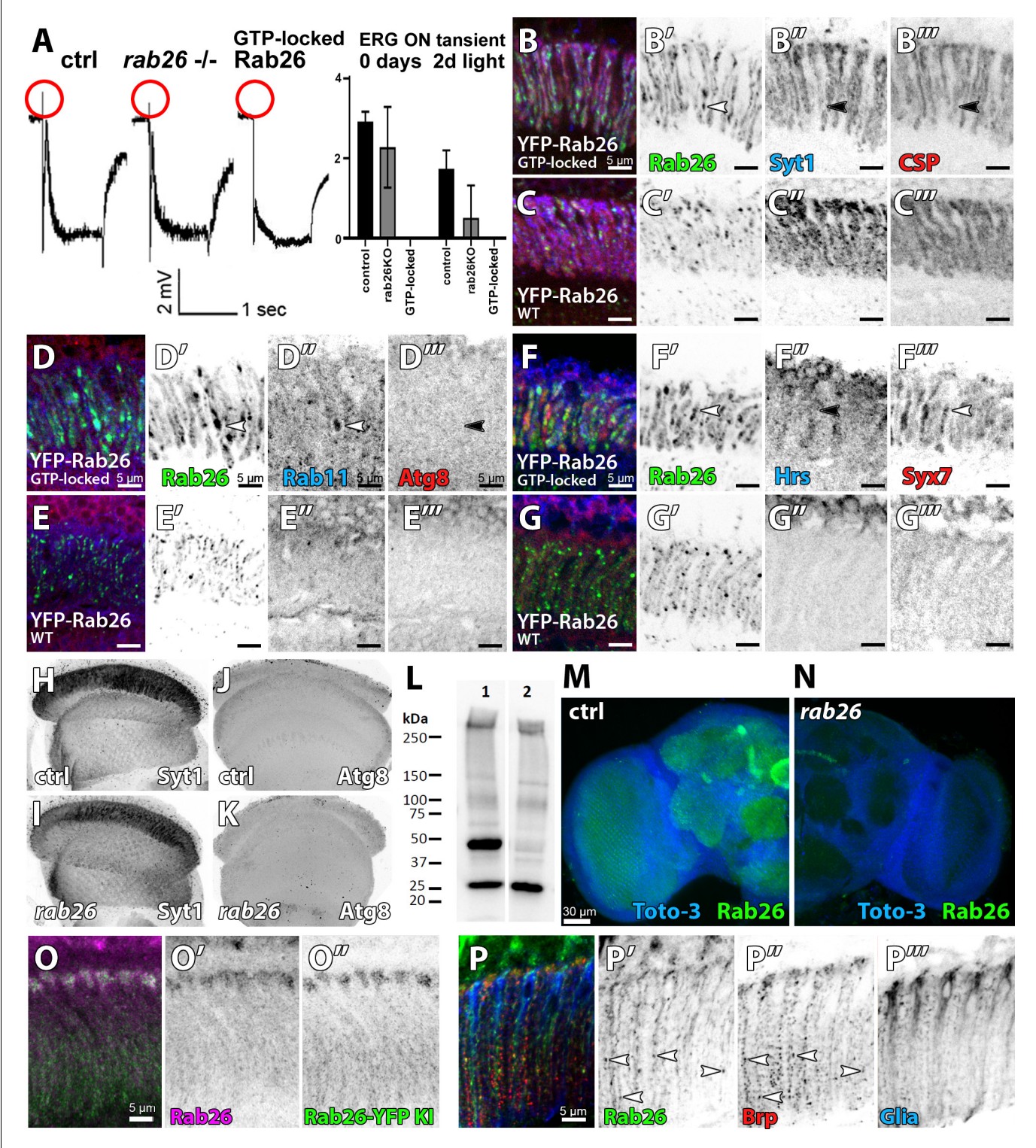

**Figure 6.** Loss of *rab26* does not discernibly affect markers for synaptic vesicles or autophagy in the adult brain. (**A**) Representative ERG traces of recordings of 2 days light stimulated wild type, *rab26* mutant, and Rab26 GTP-locked overexpression flies. Only the Rab26 GTP-locked flies show a complete loss of 'on' transient (highlighted in red). Quantification of the 'on' transient is shown right. (**B–G**) Labeling of lamina cross-sections of Rab26 GTP-locked (**B, D, and F**) and YFP-tagged Rab26WT (**C, E, and G**) against Syt1 and CSP (**B and C**), Rab11 and ATG8 (**D and E**), and Hrs and Syx7/ Avalanche (**F and G**). GTP-locked Rab26 shows colocalization with Rab11 and Syx7/Avalanche (white arrowheads), but not with Syt1, CSP, Atg8 nor Hrs (black arrowheads). Scale bar = 5 µm; number of brains n = 3–5 per antibody staining. (**H–K**) Intensity comparison of optic lobes of newly hatched wild

*Figure 6 continued on next page*

*Figure 6 continued*

type and *rab26* mutant flies, stained against Syt1 (**H and I**) and Atg8 (**J and K**). Number of brains n = 3–5 per antibody staining. (**L**) Validation of the *rab26* null mutant by Western Blot with the newly generated Rab26 antibody. Wild type control shows the Rab26 band at around 45 kDa (1), which is lost in the *rab26* mutant (2). (**M and N**) Validation of the *rab26* null mutant by immunohistochemistry with the newly generated Rab26 antibody. The Rab26 antibody labels synaptic neuropil in different regions of wild type brains (green, **M**), which is lost in the *rab26* null mutant (**N**). Labeling of nuclei/cell bodies with Toto-3 (blue). Scale bar = 30 µm; number of brains n = 3 per antibody staining. (**O**) Immunolabeling of Rab26 (red) shows high colocalization with the endogenously YFP-tagged Rab26 (green). Lamina cross-section of newly hatched flies. Scale bar = 5 µm; number of brains n = 3–5 per antibody staining. (**P**) Co-labeling of wild type lamina with Rab26 (green), Brp (synaptic marker, red), and ebony (glia marker, blue) reveals few synapses, positive for Rab26 and Brp in the proximal region of the lamina (white arrowheads, **P'** and **P''**). No colocalization between Rab26 and ebony could be observed (**P'''**). Scale bar = 5 µm; number of brains n = 3–5 per antibody staining.

The online version of this article includes the following figure supplement(s) for figure 6:

**Figure supplement 1.** Rab26 colocalizes with synaptic vesicle and endosomal markers at larval neuromuscular junction (NMJ) boutons.

Rab26 (see Materials and methods). In western blots of whole-brain homogenate, the Rab26 antibody labeled a 45 kDa band, consistent with a predicted molecular weight between 41 kDa and 45 kDa, that is lost in the null mutant (***Figure 6L***). Additionally, immunolabeling of Rab26 in the adult brain (***Figure 6M–N***) and at the larval NMJ (***Figure 6—figure supplement 1A***) is not detectable in the null mutant. At the NMJ, Rab26 is present at presynaptic boutons, but not in the postsynaptic muscle (***Figure 6—figure supplement 1A,B***). Rab26 immunolabeling colocalizes partially with Rab11, the synaptic vesicle markers CSP and Syt1 and the endosomal marker Syx7. However, none of these markers were discernibly affected in the *rab26* null mutant (***Figure 6—figure supplement 1B***). Similarly, overexpressed YFP-tagged Rab26, GDP-locked Rab26 and GTP-locked Rab26 exhibited varying levels of colocalization with synaptic vesicle and endosomal markers, but no obvious disruption of their localization or levels (***Figure 6—figure supplement 1C,E,F***). Finally, we found no effect of the *rab26* null mutant or overexpression of the three YFP-tagged Rab26 variants on the autophagosomal marker Atg8 (***Figure 6—figure supplement 1D***). We hypothesize that, as in photoreceptor neurons, Rab26 is not required for the formation of functional synapses.

In the adult brain, Rab26 immunolabeling revealed synaptic neuropils at varying levels in different regions (***Figure 6M***) and colocalized well with an endogenously tagged Rab26 (***Figure 6O***). In the lamina, Rab26 immunolabeling revealed a punctate pattern across the photoreceptor axon terminals and a row of cells just distal of the lamina (***Figure 6O,P***). Co-labeling with the glia marker ebony did not mark these cells and revealed a largely complementary pattern to Rab26 in the lamina; the synaptic marker Brp revealed a small subset of colocalizing synapses selectively in the proximal regions of the axon terminals (arrowheads in ***Figure 6P***), that is in the region where continuous stimulation led to protein accumulations (comp. ***Figure 5A***). These observations raise the question whether Rab26 functions specifically in a certain type of neuron or synapse.

## Rab26 is required for stimulus-dependent membrane receptor turnover associated with cholinergic synapses

So far, our *rab26* null mutant analyses have revealed a stimulus-dependent role in functional maintenance (***Figure 3E***) associated with membrane protein accumulations at the proximal end of photoreceptor synaptic terminals (***Figure 5A***). These mutant accumulations of the photoreceptor membrane protein Chaoptin became more pronounced with further increased (4 days light) stimulation (***Figure 7A–B***). This phenotype was mimicked by photoreceptor-specific Rab26 RNAi (***Figure 7—figure supplement 1A–E***) and rescued by photoreceptor-specific expression of Rab26 in null mutant flies (***Figure 7C–D***). These findings indicate that the stimulus-dependent membrane accumulations are a cell-autonomous phenotype in photoreceptor neurons.

To characterize the nature of these presynaptic protein accumulations, we tested a panel of markers for membrane-associated proteins (***Figure 7E–M***). Amongst these markers, in addition to Chaoptin, the protein accumulations were specifically enriched for the synaptic transmembrane cell adhesion molecule N-Cadherin (CadN) (***Figure 7E–G***). By contrast, neither the autophagosomal marker Atg8, the synaptic vesicle marker Syt1 (***Figure 7J–M***), nor the endosomal markers Rab5 and Rab7 were associated with the accumulations (***Figure 7E***). Of the endosomal markers, only Syx7 was significantly increased (***Figure 7E,H–I***). We conclude that continuous stimulation leads to the

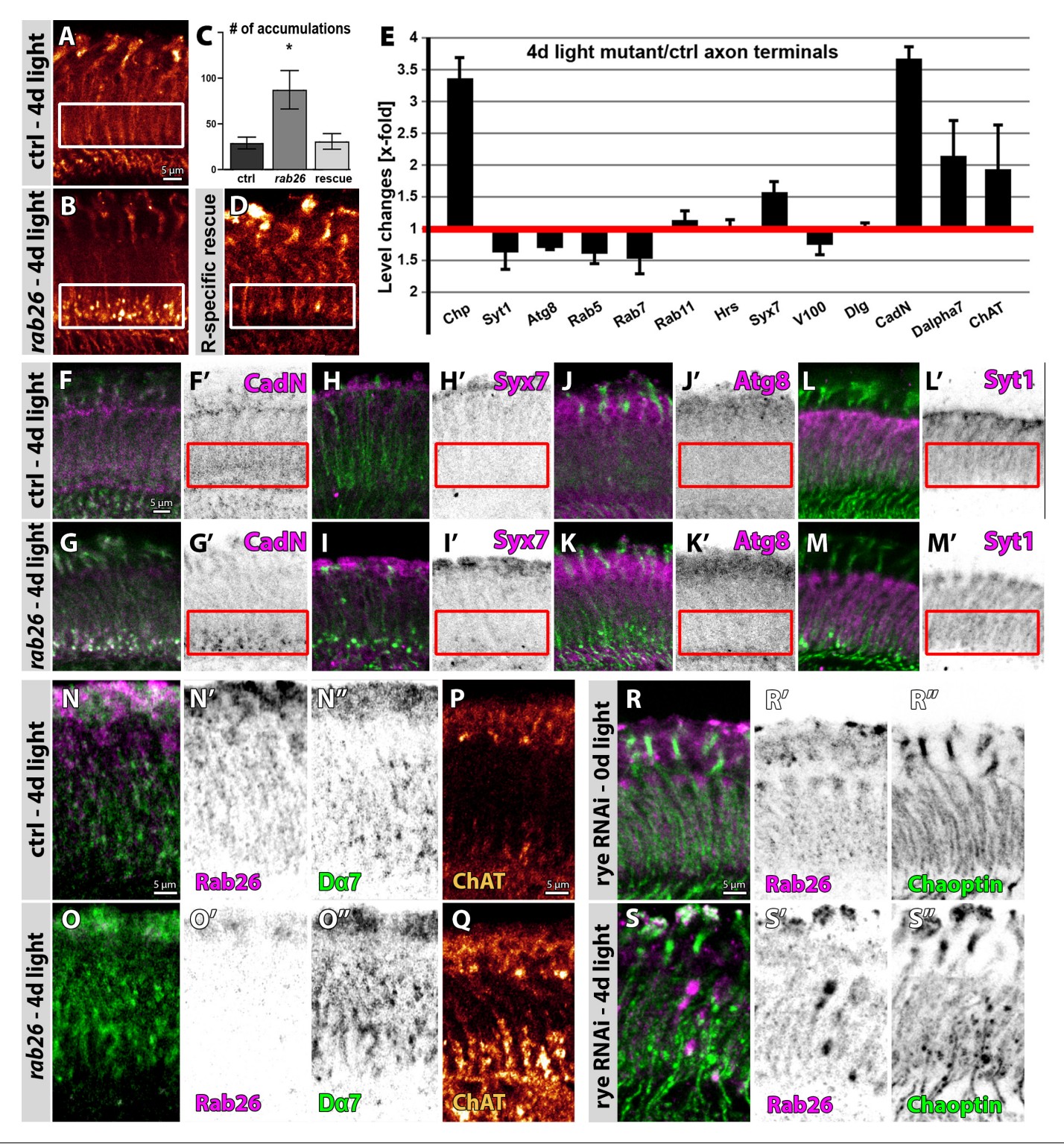

**Figure 7.** Rab26 is required for membrane receptor turnover associated with cholinergic synapses. (A–D) *rab26* mutant R1-R6 photoreceptor terminals (B) exhibit Chaoptin-positive accumulations in the proximal lamina after 4 days of light stimulation (highlighted with white boxes), which are rescued by photoreceptor-specific Rab26 expression (C and D). (C) Quantification. Mean ± SEM; *p<0.05; number of lamina per genotype n = 8; ordinary one-way ANOVA with pair-wise comparison. Scale bar = 5 µm; number of brains n = 5. (E) Quantification of level changes of 13 membrane-associated proteins in the *rab26* mutant axon terminals after 4 days of light stimulation. (F–M) Examples of lamina cross-sections of wild type (F, H, J and L) and *rab26* mutant (G, I, K and M) after 4 days of light stimulation, showing proteins that are upregulated in R1-R6 terminals (CadN, (F–G); Syx7 (H–I)) and proteins

*Figure 7 continued on next page*

*Figure 7 continued*

that are unaffected (Atg8, (**J–K**); Syt1, (**L–M**)). The proximal lamina region is highlighted by red boxes. Scale bar = 5 µm; number of brains n = 3–5 per antibody staining. (**N–O**) The *rab26* mutant exhibits an increase of Dα7 (green) across the lamina compared to wild type after 4 days of light stimulation. Shown are lamina cross-sections. Scale bar = 5 µm; number of brains n = 3–5 per antibody staining. (**P–Q**) The *rab26* mutant shows an increase of ChAT in the proximal lamina compared to wild type after 4 days of light stimulation. Scale bar = 5 µm; number of brains n = 3–5 per antibody staining. (**R–S**) Photoreceptor-specific knock down of rye leads to an increase of Chaoptin and Rab26 in the lamina after 4 days of light stimulation (**S**) compared to newly hatched flies (**R**). Rab26 accumulates throughout the lamina (**S'**), whereas Chaoptin accumulates in the proximal lamina (**S''**). Scale bar = 5 µm; number of brains n = 3–5 per antibody staining.

The online version of this article includes the following figure supplement(s) for figure 7:

**Figure supplement 1.** Rab26 RNAi recapitulates the null mutant lamina phenotype.

selective accumulation of presynaptic transmembrane receptors, including Chaoptin and CadN, specifically in the most proximal part of photoreceptor terminals.

Amongst lamina neurons, only L4 specifically forms synapses at the most proximal end of photoreceptor axon terminals (*Fischbach and Dittrich, 1989*; *Lüthy et al., 2014*; *Rivera-Alba et al., 2011*; *Tadros et al., 2016*). L4 neurons function in the detection of progressive motion (*Tuthill et al., 2013*) and are cholinergic based on the expression of the vesicular acetylcholine transporter and choline acetyltransferase (ChAT) (*Davis et al., 2020*; *Kolodziejczyk et al., 2008*). Immunolabeling of presynaptic ChAT and the postsynaptic cholinergic receptor Dα7 (*Fayyazuddin et al., 2006*) revealed increased levels of both proteins after 4 days of light stimulation, with ChAT increases specific to the proximal lamina, while Dα7 appears across the entire lamina (*Figure 7N–Q*). Across the optic lobe, the endogenous Rab26 knock-in exhibits an expression pattern similar to ChAT (*Figure 7—figure supplement 1F*). However, photoreceptors that terminate in the lamina are not known to be cholinergic, and they neither express ChAT nor the Dα7 receptor based on a recent systematic transcriptome analysis (*Davis et al., 2020*).

Amongst lamina neurons, L4 and lamina wide-field feedback (Lawf) neurons have been shown to be both cholinergic and provide synaptic input to R1-R6 photoreceptor axon terminals (*Davis et al., 2020*; *Rivera-Alba et al., 2011*). Co-labeling of these neurons with Rab26 and ChAT (*Figure 7—figure supplement 1G–I*) revealed that the Rab26-positive cells distal of the lamina were Lawf2 neurons (*Figure 7—figure supplement 1H–I*), while the ChAT-positive labeling in the proximal lamina colocalized with L4 (*Figure 7—figure supplement 1G*); Rab26 labeling was complementary to the ChAT-positive L4 collaterals (*Figure 7—figure supplement 1G*).

In addition to receiving input from cholinergic L4 neurons (*Rivera-Alba et al., 2011*), photoreceptors are predicted to express a single acetylcholine receptor subunit, Dα4 (*Davis et al., 2020*). Dα4, also called redeye (rye), was previously found to promote sleep in *Shi et al., 2014*. We therefore used an RNAi approach established in the sleep study to knock down Dα4 specifically in photoreceptor neurons. Dα4 RNAi exhibited no obvious defects prior to stimulation (*Figure 7R*). By contrast, after 4 days of light stimulation, photoreceptor-specific Dα4 RNAi led to both Rab26-positive accumulations in the lamina as well as the proximal Chaoptin accumulations characteristic for the *rab26* mutant after stimulation (*Figure 7—figure supplement 1*). Hence, loss of *rab26* in photoreceptors has a stimulus-dependent effect similar to decreased cholinergic input onto photoreceptor axon terminals, that function as postsynaptic partners in this case. These findings suggest a specialized role of Rab26 in stimulus-dependent, synapse-specific receptor trafficking.

## Discussion

In this study, we generated a complete *rab* null mutant collection and provide comparative functional analyses of those that are viable under laboratory conditions. Surprisingly, all previously described nervous system-enriched Rab GTPases fall into this category. However, challenging development with temperature or challenging function with continuous stimulation revealed distinct requirements for all homozygous viable *rabs*. Our findings suggest that the majority of Rab GTPases modulate membrane trafficking in neurons and other tissues to maintain robust development and function under challenging environmental conditions.

## A functional *rab* family profile

Since the identification of Ypt1 (Rab1) in yeast, the Rab GTPase family has been well characterized as an evolutionarily conserved group of proteins involved in the regulation of membrane trafficking in all eukaryotes (*Hutagalung and Novick, 2011*; *Klöpper et al., 2012*; *Lipatova et al., 2015*; *Pfeffer, 2017*). Rab GTPases have been analyzed in several comparative studies in order to gain a systematic view of membrane trafficking in cells (*Best and Leptin, 2020*; *Chan et al., 2011*; *Dunst et al., 2015*; *Gillingham et al., 2014*; *Gurkan et al., 2005*; *Harris and Littleton, 2011*; *Jin et al., 2012*; *Pfeffer, 1994*; *Stenmark, 2009*; *Zerial and McBride, 2001*). All comparative studies to date have been based on expression profiling, the expression of GDP- and GTP-locked Rabs or RNAi. As a cautionary note, we have previously described differences between loss of gene function and the expression of GDP-locked (often called dominant negative) variants (*Chan et al., 2011*; *Cherry et al., 2013*). The complete mutant collection allows the comparison of molecularly defined null mutants with other functional perturbation approaches for all 26 *Drosophila rab* genes.

The *Drosophila rab* null mutant collection and comparative characterization of all viable *rabs* provides an opportunity for a comprehensive comparison of the Rab family between *Drosophila* and other species. We have therefore assembled available information on viability, function, subcellular localization and expression patterns for all Rabs in several mammalian species, flies and yeast. *Supplementary file 3* provides a comparison of functional and subcellular localization data for Rabs in different mammals, *D. melanogaster* and *S. cerevisiae*. Amongst a wealth of information in phenotypic homologies, these data also show that the majority of Rab family members yield viable organisms under laboratory conditions when mutated. *Supplementary file 2* provides a comparison of differential tissue expression in multicellular animal species. These data reveal numerous parallels especially with respect to enrichment in the nervous system. Rabs are listed according to lineage tracing and homology pairing, as comprehensively reported previously (*Hutagalung and Novick, 2011*; *Klöpper et al., 2012*; *Pereira-Leal and Seabra, 2000*; *Pereira-Leal and Seabra, 2001*; *Zhang et al., 2007*).

Our mutant analyses highlight that viability vs lethality is not a binary distinction of the null mutants, but represents a continuous range of context-dependent phenotypes (*Hiesinger, 2021*). Of the 26 null mutants, only seven are fully lethal under laboratory conditions in our study (*rab1*, *rab2*, *rab5*, *rab6*, *rab7*, *rab8*, *rab11*), while an eighth mutant is 'semi-lethal' based on few adult escapers (*rab35*). Two more lines are viable, but infertile as homozygous adults (*rab10*, *rab30*). Several others are highly sensitive to rearing conditions and may appear lethal depending on for example temperature, including *rabX1*, *rabX4*, *rab19*, and *rab32*. In addition, several mutants exhibit reduced numbers of offspring or developmental or neuronal functional impairments depending on environmental conditions. Similar sensitivities and reduced viability have been found for several mammalian *rabs* (*Supplementary file 3*).

Based on an analysis of endogenously tagged Rabs (*Dunst et al., 2015*), all 13 nervous system Rabs are expressed in varying patterns in the nervous system with predominant protein localization to synaptic neuropils (*Figure 1—figure supplements 2–3*; *Supplementary file 1*), consistent with our previous analyses of tagged Rabs in the larval nervous system (*Chan et al., 2011*; *Jin et al., 2012*). All mutants with stimulus-dependent functional maintenance defects exhibit strong adult synaptic localization (*Table 1*). These observations support the idea that the majority of Rabs with adult synaptic localization serve modulatory functions that become apparent under light challenging conditions, namely Rab3, Rab26, Rab19, RabX6, Rab30, and RabX4. By contrast, Rab27, Rab32, Rab23, and Rab9 are more likely to serve cell-specific functions, consistent with previous observations for each of the four in *Drosophila* (*Chan et al., 2011*; *Dong et al., 2013*; *Gillingham et al., 2014*; *Lien et al., 2020*; *Ma et al., 2004*).

## Neuronal maintenance, membrane trafficking, and the role of *rab26*

Our previous systematic analysis was based on expression profiling and suggested that the nervous system exhibits particularly pronounced expression of all Rab GTPases in *Drosophila* (*Chan et al., 2011*; *Jin et al., 2012*). We were surprised to find that all Rabs identified to be particularly enriched in the nervous system proved to be viable under laboratory conditions. However, laboratory conditions avoid environmental challenges while nervous system development and function have evolved robustness to variable conditions (*Hiesinger and Hassan, 2018*).

**Table 1.** Summary of functional analyses.

| | Viability and development | | | | | Temp. sens. | | Neuronal function | | | | | | | |
|---|---|---|---|---|---|---|---|---|---|---|---|---|---|---|---|
| | Viability | Total dev. | Embryo | Larva | Pupa | Lethal | Wing | Syn 2d | Depol 2d | Syn dark | Depol dark | Rhabd. 2d | Axon morph | Rab11 | Atg8 |
| Rab3 | | only 18°C | | only 18°C | | | 18°C | | | | | | | | |
| RabX4 | Reduced | | | | | 18°C | 18°C | | | | | | | | |
| Rab27 | | | | | | | 18°C | | | | | Area | | | |
| Rab26 | | | | | | | | | | | | Shape | | | |
| Rab19 | | | | | | 29°C | | | | | | Shape | | | |
| Rab32 | Reduced | only 18°C | | only 18°C | | 29°C | | | | | | Area | | | |
| RabX1 | Reduced | | | | | | 18°C | | | | | | | | |
| RabX6 | | only 18°C | | | | | 29°C | | | | | | | | |
| Rab40 | Reduced | | | only 18°C | | | | | | | | Area | | | |
| Rab23 | Reduced | | | | | | | | | | | Shape | | | |
| Rab21 | | | | | | | | | | | | Area | | | |
| Rab9 | | | | | | 29°C | | | | | | | | | |
| Rab4 | | | | only 18°C | only 18°C | | | | | | | Area | | | |
| Rab14 | Reduced | | | | | | | | | | | | | | |
| Rab39 | | only 18°C | | | | | | | | | | | | | |
| Rab18 | | | | | | | | | | | | Area | | | |
| Rab10 | Infertile | | | | | | 18°C | | | | | | | | |
| Rab30 | Infertile | | | | | | | | | | | Area | | | |
| Rab7 | Lethal | | | | | | | | | | | | | | |
| Rab8 | Lethal | | | | | | | | | | | | | | |
| Rab2 | Lethal | | | | | | | | | | | | | | |
| Rab1 | Lethal | | | | | | | | | | | | | | |
| Rab6 | Lethal | | | | | | | | | | | | | | |
| Rab35 | Semi-lethal | | | | | | | | | | | | | | |
| Rab5 | Lethal | | | | | | | | | | | | | | |
| Rab11 | Lethal | | | | | | | | | | | | | | |

Overview of analyses ('Viability and Development', 'Temperature sensitivity' and 'Neuronal Function') done in this study for the indicated Rab GTPases. Abbreviations: bc = backcrossed *rab* mutants, depol = depolarization, dev. = development, Df = deficiency, morph = morphology, Rhabdom = rhabdomere, sens = sensitivity, syn = synaptic, temp = temperature, 2d = 2 days.

Color code: green denotes no difference to control; grey through yellow and orange denotes increasing deviation from controls in functional analyses.

It is likely that key roles of Rab-dependent functions are executed by the lethal mutants not analyzed here. For example, *rab7* is a ubiquitously expressed gene, but disease-associated mutations primarily affect the nervous system and cause the neuropathy CMT2B (*Cherry et al., 2013*; *Verhoeven et al., 2003*). In axon terminals, local *rab7*-dependent degradation is required for turnover of membrane receptors, but not synaptic vesicles (*Jin et al., 2018b*). While null mutants for *rab7* are lethal, haploinsufficiency revealed neuronal sensitivity to reduced membrane degradation (*Cherry et al., 2013*). Similar to heterozygous *rab7*, our analyses of viable lines suggest that such evolutionarily selected functional properties may 'hide' in mutants that are characterized as viable under laboratory conditions.

Neurons require compartment-specific membrane trafficking in both axon terminals and dendrites (*Jin et al., 2018a*; *Jin et al., 2018b*). At presynaptic axon terminals, Rabs have been implicated in synaptic vesicle recycling, synaptic development and maintenance (*Binotti et al., 2015*; *Graf et al., 2009*; *Sheehan et al., 2016*; *Uytterhoeven et al., 2011*). We previously found that several neuron-enriched Rabs at axon terminals were positive for the recycling endosome marker Rab11

(*Chan et al., 2011*), including Rab26. Rab26 was subsequently identified as a possible link between autophagy and synaptic vesicle recycling (*Binotti et al., 2015*). Here, we describe that *rab26* mutants indeed exhibited neuronal functional defects when challenged with continuous stimulation. However, we did not find obvious changes to autophagosomal and synaptic vesicle markers in the null mutant. Instead, the null mutant revealed stimulation-dependent increases of selected membrane proteins, including the presynaptic choline acetyltransferase (ChAT) and the postsynaptic alpha7 acetylcholine receptor. Correspondingly, the Rab26 protein is highly enriched in cholinergic neurons in the fly visual system. Interestingly, R1-R6 photoreceptors are not cholinergic, but are predicted to express the acetylcholine receptor alpha4 (*Davis et al., 2020*). Our findings support an unusual postsynaptic role of the R1-R6 axon terminals for cholinergic, Rab26-dependent signaling from L4 neurons through feedback synapses (*Rivera-Alba et al., 2011*). We speculate that these feedback synapses are activated by continuous visual stimulation and lead to Rab26-dependent receptor endocytosis defects in the photoreceptor terminals. Based on this idea, it will be interesting to test the role of Rab26 at other cholinergic synapses and test its requirement in an activity-dependent manner. We conclude that the study of *rab* mutants that are viable under laboratory conditions may help to elucidate an understanding of evolutionarily selected functional requirements of the nervous system under varying environmental conditions. The complete collection of null mutants provides a resource designed to facilitate such further studies.

# Materials and methods

## Key resources table

| Reagent type (species) or resource | Designation | Source or reference | Identifiers | Additional information |
|---|---|---|---|---|
| Gene (*D. melanogaster*) | Rab2 | | FlyBase ID:FBgn0014009 | Sequence location: 2R:6,696,739.6,699,469 [+] |
| Gene (*D. melanogaster*) | Rab4 | | FlyBase ID:FBgn0016701 | Sequence location: 2R:17,573,462.17,574,979 [+] |
| Gene (*D. melanogaster*) | Rab9 | | FlyBase ID:FBgn0032782 | Sequence location: 2L:19,432,574.19,435,841 [+] |
| Gene (*D. melanogaster*) | Rab10 | | FlyBase ID:FBgn0015789 | Sequence location: X:20,251,338.20,254,691 [+] |
| Gene (*D. melanogaster*) | Rab14 | | FlyBase ID:FBgn0015791 | Sequence location: 2L:14,355,145.14,358,764 [+] |
| Gene (*D. melanogaster*) | Rab18 | | FlyBase ID:FBgn0015794 | Sequence location: X:5,670,827.5,671,812 [-] |
| Gene (*D. melanogaster*) | Rab19 | | FlyBase ID:FBgn0015793 | Sequence location: 3L:8,297,018.8,298,506 [+] |
| Gene (*D. melanogaster*) | Rab21 | | FlyBase ID:FBgn0039966 | Sequence location: X:23,012,140.23,013,409 [-] |
| Gene (*D. melanogaster*) | Rab23 | | FlyBase ID:FBgn0037364 | Sequence location: 3R:5,680,054.5,685,434 [-] |
| Gene (*D. melanogaster*) | Rab26 | | FlyBase ID:FBgn0086913 | Sequence location: 3L:21,318,774.21,335,027 [+] |
| Gene (*D. melanogaster*) | Rab30 | | FlyBase ID:FBgn0031882 | Sequence location: 2L:7,030,493.7,032,606 [-] |
| Gene (*D. melanogaster*) | Rab35 | | FlyBase ID:FBgn0031090 | Sequence location: X:20,155,766.20,159,872 [-] |
| Gene (*D. melanogaster*) | Rab39 | | FlyBase ID:FBgn0029959 | Sequence location: X:7,734,923.7,736,756 [+] |
| Gene (*D. melanogaster*) | Rab40 | | FlyBase ID:FBgn0030391 | Sequence location: X:12,459,796.12,463,112 [-] |
| Gene (*D. melanogaster*) | RabX1 | | FlyBase ID:FBgn0015372 | Sequence location: 2R:23,519,839.23,523,613 [-] |

*Continued on next page*

Continued

| Reagent type (species) or resource | Designation | Source or reference | Identifiers | Additional information |
|---|---|---|---|---|
| Gene (*D. melanogaster*) | RabX4 | | FlyBase ID:FBgn0051118 | Sequence location: 3R:24,826,665.24,828,409 [-] |
| Gene (*D. melanogaster*) | RabX6 | | FlyBase ID: FBgn0035155 | Sequence location: 3L:690,517.691,951 [+] |
| Strain, strain background (*D. melanogaster*) | yw | | | yw;; |
| Strain, strain background (*D. melanogaster*) | w$^{1118}$ | | | w$^{1118}$;; |
| Genetic reagent (*D. melanogaster*) | rab30$^{-Gal4-KI}$, UAS-YFP-Rab30WT | Hiesinger lab stock | | |
| Genetic reagent (*D. melanogaster*) | rab3-Df | Bloomington *Drosophila* Stock Center (BDSC) | BDSC:8909 | Deficiency line for rab3 |
| Genetic reagent (*D. melanogaster*) | rab4-Df | Bloomington *Drosophila* Stock Center | BDSC:38465 | Deficiency line for rab4 |
| Genetic reagent (*D. melanogaster*) | rab9-Df | Bloomington *Drosophila* Stock Center | BDSC:7849 | Deficiency line for rab9 |
| Genetic reagent (*D. melanogaster*) | rab10-Df | Bloomington *Drosophila* Stock Center | BDSC:29995 | Deficiency line for rab10 |
| Genetic reagent (*D. melanogaster*) | rab14-Df | Bloomington *Drosophila* Stock Center | BDSC:7518 | Deficiency line for rab14 |
| Genetic reagent (*D. melanogaster*) | rab19-Df | Bloomington *Drosophila* Stock Center | BDSC:7591 | Deficiency line for rab19 |
| Genetic reagent (*D. melanogaster*) | rab32-Df | Bloomington *Drosophila* Stock Center | BDSC:23664 | Deficiency line for rab32 |
| Genetic reagent (*D. melanogaster*) | rab39-Df | Bloomington *Drosophila* Stock Center | BDSC:26563 | Deficiency line for rab39 |
| Genetic reagent (*D. melanogaster*) | rab40-Df | Bloomington *Drosophila* Stock Center | BDSC:26578 | Deficiency line for rab40 |
| Genetic reagent (*D. melanogaster*) | rabX1-Df | Bloomington *Drosophila* Stock Center | BDSC:26513 | Deficiency line for rabX1 |
| Genetic reagent (*D. melanogaster*) | rabX4-Df | Bloomington *Drosophila* Stock Center | BDSC:25024 | Deficiency line for rabX4 |
| Genetic reagent (*D. melanogaster*) | rabX6-Df | Bloomington *Drosophila* Stock Center | BDSC:8048 | Deficiency line for rabX6 |
| Genetic reagent (*D. melanogaster*) | EYFP-Rab3 | ***Dunst et al., 2015*** | FlyBase ID:FBst0 062541; BDSC:62541 | FlyBase Genotype: w$^{1118}$; TI{TI}Rab3$^{EYFP}$ |
| Genetic reagent (*D. melanogaster*) | EYFP-Rab4 | ***Dunst et al., 2015*** | FlyBase ID:FBst0062542; BDSC:62542 | FlyBase Genotype: y$^1$ w$^{1118}$; TI{TI}Rab4$^{EYFP}$ |
| Genetic reagent (*D. melanogaster*) | EYFP-Rab9 | ***Dunst et al., 2015*** | FlyBase ID:FBst0062547; BDSC:62547 | FlyBase Genotype: w$^{1118}$; TI{TI}Rab9$^{EYFP}$ |
| Genetic reagent (*D. melanogaster*) | EYFP-Rab19 | ***Dunst et al., 2015*** | FlyBase ID:FBst0062552; BDSC:62552 | FlyBase Genotype: w$^{1118}$; TI{TI}Rab19$^{EYFP}$ |

*Continued*

| Reagent type (species) or resource | Designation | Source or reference | Identifiers | Additional information |
|---|---|---|---|---|
| Genetic reagent (*D. melanogaster*) | EYFP-Rab21 | *Dunst et al., 2015* | FlyBase ID:FBst0062553; BDSC:62553 | FlyBase Genotype: $y^1$ $w^{1118}$ TI{TI}Rab21$^{EYFP}$ |
| Genetic reagent (*D. melanogaster*) | EYFP-Rab23 | *Dunst et al., 2015* | FlyBase ID:FBst0062554; BDSC:62554 | FlyBase Genotype: $y^1$ $w^{1118}$; TI{TI}Rab23$^{EYFP}$ |
| Genetic reagent (*D. melanogaster*) | EYFP-Rab26 | *Dunst et al., 2015* | FlyBase ID:FBst0062555; BDSC:62555 | FlyBase Genotype: $y^1$ $w^{1118}$; TI{TI}Rab26$^{EYFP}$ |
| Genetic reagent (*D. melanogaster*) | EYFP-Rab27 | *Dunst et al., 2015* | FlyBase ID:FBst0062556; BDSC:62556 | FlyBase Genotype: $y^1$ TI{TI}Rab27$^{EYFP}$ $w^{1118}$ |
| Genetic reagent (*D. melanogaster*) | EYFP-Rab32 | *Dunst et al., 2015* | FlyBase ID:FBst0062558; BDSC:62558 | FlyBase Genotype: $w^{1118}$; TI{TI}Rab32$^{EYFP}$ |
| Genetic reagent (*D. melanogaster*) | EYFP-Rab40 | *Dunst et al., 2015* | FlyBase ID:FBst0062561; BDSC:62561 | FlyBase Genotype: $y^1$ $w^{1118}$ TI{TI}Rab40$^{EYFP}$ |
| Genetic reagent (*D. melanogaster*) | EYFP-RabX1 | *Dunst et al., 2015* | FlyBase ID:FBst0062562; BDSC:62562 | FlyBase Genotype: $w^{1118}$; TI{TI}RabX1$^{EYFP}$ |
| Genetic reagent (*D. melanogaster*) | EYFP-RabX4 | *Dunst et al., 2015* | FlyBase ID:FBst0062563; BDSC:62563 | Heterozygous flies used; FlyBase Genotype: $w^{1118}$; TI{TI}RabX4$^{EYFP}$ |
| Genetic reagent (*D. melanogaster*) | EYFP-RabX6 | *Dunst et al., 2015* | FlyBase ID:FBst0062565; BDSC:62565 | FlyBase Genotype: $w^{1118}$; TI{TI}RabX6$^{EYFP}$ |
| Genetic reagent (*D. melanogaster*) | rab2 | This paper | | Fly stock maintained in Hiesinger lab; see Materials and methods |
| Genetic reagent (*D. melanogaster*) | rab4 | This paper | | Fly stock maintained in Hiesinger lab; see Materials and methods |
| Genetic reagent (*D. melanogaster*) | rab9 | This paper | | Fly stock maintained in Hiesinger lab; see Materials and methods |
| Genetic reagent (*D. melanogaster*) | rab10 | This paper | | Fly stock maintained in Hiesinger lab; see Materials and methods |
| Genetic reagent (*D. melanogaster*) | rab14 | This paper | | Fly stock maintained in Hiesinger lab; see Materials and methods |
| Genetic reagent (*D. melanogaster*) | rab18 | This paper | | Fly stock maintained in Hiesinger lab; see Materials and methods |
| Genetic reagent (*D. melanogaster*) | rab19 | This paper | | Fly stock maintained in Hiesinger lab; see Materials and methods |
| Genetic reagent (*D. melanogaster*) | rab21 | This paper | | Fly stock maintained in Hiesinger lab; see Materials and methods |
| Genetic reagent (*D. melanogaster*) | rab23 | This paper | | Fly stock maintained in Hiesinger lab; see Materials and methods |
| Genetic reagent (*D. melanogaster*) | rab26 | This paper | | Fly stock maintained in Hiesinger lab; see Materials and methods |
| Genetic reagent (*D. melanogaster*) | rab30 | This paper | | Fly stock maintained in Hiesinger lab; see Materials and methods |

*Continued on next page*

*Continued*

| Reagent type (species) or resource | Designation | Source or reference | Identifiers | Additional information |
|---|---|---|---|---|
| Genetic reagent (*D. melanogaster*) | *rab35* | This paper | | Fly stock maintained in Hiesinger lab; see Materials and methods |
| Genetic reagent (*D. melanogaster*) | *rab39* | This paper | | Fly stock maintained in Hiesinger lab; see Materials and methods |
| Genetic reagent (*D. melanogaster*) | *rab40* | This paper | | Fly stock maintained in Hiesinger lab; see Materials and methods |
| Genetic reagent (*D. melanogaster*) | *rabX1* | This paper | | Fly stock maintained in Hiesinger lab; see Materials and methods |
| Genetic reagent (*D. melanogaster*) | *rabX4* | This paper | | Fly stock maintained in Hiesinger lab; see Materials and methods |
| genetic reagent (*D. melanogaster*) | *rabX6* | This paper | | Fly stock maintained in Hiesinger lab; see Materials and methods |
| Genetic reagent (*D. melanogaster*) | *rab1* | *Thibault et al., 2004* | FlyBase ID:FBst0017936; BDSC:17936 | FlyBase Genotype: $w^{1118}$; PBac{RB}Rab1$^{e01287}$/TM6B, Tb$^1$ |
| Genetic reagent (*D. melanogaster*) | *rab3* | *Graf et al., 2009* | FlyBase ID:FBst0078045; BDSC:78045 | FlyBase Genotype: $w^*$; Rab3$^{rup}$ |
| Genetic reagent (*D. melanogaster*) | *rab5* | *Wucherpfennig et al., 2003* | FlyBase ID:FBal0182047 | w; Rab5$^2$ P{neoFRT}40A/CyO; |
| Genetic reagent (*D. melanogaster*) | *rab6* | *Purcell and Artavanis-Tsakonas, 1999* | FlyBase ID: FBst0005821; BDSC:5821 | FlyBase Genotype: w*; Rab6$^{D23D}$/CyO; ry$^{506}$ |
| Genetic reagent (*D. melanogaster*) | *rab7* | *Cherry et al., 2013* | FlyBase ID:FBal0294205 | Fly stock maintained in Hiesinger lab; ";Sp/CyO; P{neoFRT}82B, Rab7$^{Gal4-KO}$ /TM3' |
| Genetic reagent (*D. melanogaster*) | *rab8* | *Giagtzoglou et al., 2012* | FlyBase ID:FBst0026173; BDSC:26173 | FlyBase Genotype: Rab8$^1$ red$^1$ e$^4$/TM6B, Sb$^1$ Tb$^1$ ca$^1$ |
| Genetic reagent (*D. melanogaster*) | *rab11* | *Bellen et al., 2004* | FlyBase ID:FBst0042708; BDSC:42708 | FlyBase Genotype: w$^*$; P{EP}Rab11$^{EP3017}$/TM6B, Tb$^1$ |
| Genetic reagent (*D. melanogaster*) | *rab27* | *Chan et al., 2011* | | Fly stock maintained in Hiesinger lab; rab27$^{Gal4-KO}$;; |
| Genetic reagent (*D. melanogaster*) | *rab32* | *Ma et al., 2004* | FlyBase ID:FBst0000338; BDSC:338 | FlyBase Genotype: Rab32$^1$ |
| Genetic reagent (*D. melanogaster*) | lGMR-Gal4, UAS-white RNAi | Hiesinger lab stock | | Fly stock maintained in Hiesinger lab; long version of GMR |
| Genetic reagent (*D. melanogaster*) | UAS-YFP-Rab26WT | *Zhang et al., 2007* | BDSC:23245 | YFP-tagged, wild type form of Rab26 |
| Genetic reagent (*D. melanogaster*) | UAS-YFP-Rab26CA | *Zhang et al., 2007* | BDSC:9809 | YFP-tagged, constitutively active form of Rab26 |
| Genetic reagent (*D. melanogaster*) | UAS-YFP-Rab26DN | *Zhang et al., 2007* | BDSC:9807 | YFP-tagged, dominant negative form of Rab26 |
| Genetic reagent (*D. melanogaster*) | elav-Gal4 | Bloomington *Drosophila* Stock Center | FlyBase ID:FBst0008765; BDSC:8765 | FlyBase Genotype: P{GAL4-elav.L}2/CyO |
| Genetic reagent (*D. melanogaster*) | sGMR-Gal4 | Bloomington *Drosophila* Stock Center | FlyBase ID:FBst0001104; BDSC:1104 | FlyBase Genotype: w$^*$; P{GAL4-ninaE.GMR}12 |

*Continued on next page*

*Continued*

| Reagent type (species) or resource | Designation | Source or reference | Identifiers | Additional information |
|---|---|---|---|---|
| Genetic reagent (*D. melanogaster*) | UAS-Rab26 RNAi | Vienna *Drosophila* Resource Center (VDRC) | VDRC:101330 | Rab26 RNAi line KK107584 |
| Genetic reagent (*D. melanogaster*) | *rab26*$^{exon1}$-Gal4 | *Chan et al., 2011* | | Fly stock is maintained in Hiesinger lab |
| Genetic reagent (*D. melanogaster*) | UAS-CD4-tdGFP | Bloomington *Drosophila* Stock Center | FlyBase ID:FBst0035839; BDSC:35839 | FlyBase Genotype: y$^1$ w$^*$; P{UAS-CD4-tdGFP}8 M2 |
| Genetic reagent (*D. melanogaster*) | 31C06-Gal4 (L4-Gal4) | Bloomington *Drosophila* Stock Center | FlyBase ID: FBst0049883; BDSC:49883 | FlyBase Genotype: w$^{1118}$; P{GMR31C06-GAL4}attP2 |
| Genetic reagent (*D. melanogaster*) | Lawf1-Split-Gal | *Tuthill et al., 2013* | | R11G01AD attP40; R17C11DBD attP2; 'SS00772' |
| Genetic reagent (*D. melanogaster*) | Lawf2-Split-Gal | *Tuthill et al., 2013* | | R11D03AD attP40; R19C10DBD attP2; 'SS00698' |
| Genetic reagent (*D. melanogaster*) | UAS-rye RNAi; UAS-Dicer2 | Gift from Amita Sehgal | | Dα4 receptor subunit RNAi line |
| Genetic reagent (*D. melanogaster*) | *rdgC*$^{306}$ | Bloomington *Drosophila* Stock Center | FlyBase ID:FBst0003601; BDSC:3601 | FlyBase Genotype: w$^{1118}$; rdgC$^{306}$ kar$^1$ ry$^1$/TM3, Sb$^1$ Ser$^1$ |
| Antibody | Anti-Rab5 (Rabbit polyclonal) | Abcam (Cambridge, UK) | Cat #: ab31261; RRID: AB_882240 | IHC (1:1000) |
| Antibody | Anti-Rab7 (Rabbit polyclonal) | Gift from Patrick Dolph | | IHC (1:1000) |
| Antibody | Anti-Rab11 (Mouse monoclonal) | BD Biosciences (San Jose, CA, USA) | clone47; RRID:AB_397983 | IHC (1:500) |
| Antibody | Anti-Rab26 (Guinea pig polyclonal) | This paper | | See Materials and methods; IHC (1:2000); WB (1:1000) |
| Antibody | Anti-Syt1 (Mouse monoclonal) | Developmental Studies Hybridoma Bank (DSHB) (Iowa City, IA, USA) | 3H2 2D7; RRID:AB_528483 | IHC (1:500) |
| Antibody | Anti-GABARAP+ GABARAPL1+ GABARAPL2 (Atg8) (Rabbit monoclonal) | Abcam (Cambridge, UK) | Cat #: ab109364; RRID:AB_10861928 | IHC (1:100) |
| Antibody | Anti-Syx7/Avalanche (Rabbit polyclonal) | Gift from Helmut Kramer | | IHC (1:1000) |
| Antibody | Anti-Hrs (Guinea pig polyclonal) | Gift from Hugo Bellen | | IHC (1:300) |
| Antibody | Anti-HRP (Rabbit polyclonal) | Jackson Immuno Research Laboratories (West Grove, PA, USA) | RRID:AB_2314648 | IHC (1:500) |
| Antibody | Anti-DPAK (Rabbit polyclonal) | | | IHC (1:2000) |
| Antibody | Anti-Dα7 (Rat polyclonal) | Gift from Hugo Bellen | | IHC (1:2000) |
| Antibody | Anti-nCadherin (Rat monoclonal) | Developmental Studies Hybridoma Bank (DSHB) (Iowa City, IA, USA) | DN-Ex #8; RRID:AB_528121 | IHC (1:100) |
| Antibody | Anti-V100 (Guinea pig polyclonal) | *Hiesinger et al., 2005* | | IHC (1:1000) |

*Continued on next page*

*Continued*

| Reagent type (species) or resource | Designation | Source or reference | Identifiers | Additional information |
|---|---|---|---|---|
| Antibody | Anti-CSP (Mouse monoclonal) | Developmental Studies Hybridoma Bank (DSHB) (Iowa City, IA, USA) | DCSP-2 (6D6); RRID:AB_528183 | IHC (1:50) |
| Antibody | Anti-ChAT (Mouse monoclonal) | Developmental Studies Hybridoma Bank (DSHB) (Iowa City, IA, USA) | ChAT4B1; RRID:AB_528122 | IHC (1:100) |
| Antibody | Anti-nc82 (Mouse monoclonal) | Developmental Studies Hybridoma Bank (DSHB) (Iowa City, IA, USA) | RRID: AB_2314866 | IHC (1:20) |
| Antibody | Anti-ebony (Rabbit polyclonal) | | | IHC (1:200) |
| Antibody | Anti-Chaoptin (Mouse monoclonal) | Developmental Studies Hybridoma Bank (DSHB) (Iowa City, IA, USA) | 24B10; RRID: AB_528161 | IHC (1:50) |
| Antibody | Anti-DCP-1 (Rabbit polyclonal) | Cell Signaling Technology (Danvers, MA, USA) | Asp216; Cat#: 9578; RRID:AB_2721060 | IHC (1:100) |
| Antibody | DyLight 405 AffiniPure Donkey Anti-Mouse igG (H+L) | Jackson Immuno Research (West Grove, PA, USA) | 715-475-150; RRID:AB_2340839 | IHC (1:500) |
| Antibody | Alexa Fluor 488 AffiniPure Goat Anti-Mouse IgG (H+L) | Jackson Immuno Research (West Grove, PA, USA) | 115-545-003; RRID: AB_2338840 | IHC (1:500) |
| Antibody | Alexa Fluor 488 AffiniPure Goat Anti-Mouse IgG (H+L) | Jackson Immuno Research (West Grove, PA, USA) | 115-545-166; RRID: AB_2338852 | Minimal cross-reactive; IHC (1:500) |
| Antibody | Alexa Fluor 488 Affini Pure Goat Anti-Rat IgG (H+L) | Jackson Immuno Research (West Grove, PA, USA) | 112-545-167; RRID: AB_2338362 | Minimal cross-reactive; IHC (1:500) |
| Antibody | Alexa Fluor 488 Affini Pure Goat Anti-Guinea Pig IgG (H+L) | Jackson Immuno Research (West Grove, PA, USA) | 106-545-003; RRID: AB_2337438 | IHC (1:500) |
| Antibody | Cy3 AffiniPure Goat Anti-Rabbit IgG (H+L) | Jackson Immuno Research (West Grove, PA, USA) | 111-165-003; RRID: AB_2338000 | IHC (1:500) |
| Antibody | Alexa Fluor 647 Affini Pure Goat Anti-Rabbit IgG (H+L) | Jackson Immuno Research (West Grove, PA, USA) | 111-605-045; RRID: AB_2338075 | IHC (1:500) |
| Antibody | Alexa Fluor 647 AffiniPure Goat Anti-Rat IgG (H +L) | Jackson Immuno Research (West Grove, PA, USA) | 112-605-003; RRID: AB_2338393 | IHC (1:500) |
| Antibody | Goat Anti-Guinea pig IgG H&L (Cy5) | Abcam (Cambridge, UK) | Cat. #: ab102372; RRID: AB_10710629 | IHC (1:500) |
| Antibody | Cy5 AffiniPure Goat Anti -Mouse IgG (H+L) | Jackson Immuno Research (West Grove, PA, USA) | 115-175-166; RRID: AB_2338714 | Minimal cross-reactive; IHC (1:500) |
| Antibody | Cy5 AffiniPure Goat Anti-Rat IgG (H+L) | Jackson Immuno Research (West Grove, PA, USA) | 112-175-167; RRID: AB_2338264 | Minimal cross-reactive; IHC (1:500) |

*Continued on next page*

*Continued*

| Reagent type (species) or resource | Designation | Source or reference | Identifiers | Additional information |
|---|---|---|---|---|
| Antibody | Peroxidase AffiniPure Goat Anti-Guinea Pig IgG (H+L) | Jackson Immuno Research (West Grove, PA, USA) | 106-035-003; RRID: AB_2337402 | WB (1:5000) |
| Sequence-based reagent | *rab2* | This paper | PCR primers | Fwd: 5'-TGGCCACACTGTCGC TAGCC; Rev: 5'-CGCCTCCTCTACG TTGGCAG |
| Sequence-based reagent | *rab3* | This paper | PCR primers | Fwd: 5'-ACACTGAGGCGAGC TTACGC; Rev: 5'-CTACTACCGAGGAGC-GATGGG |
| Sequence-based reagent | *rab4* | This paper | PCR primers | Fwd: 5'- GGTTTTGATCGTGTCC TGCG; Rev: 5'-AGACAACTCTTACCGC TGCC |
| Sequence-based reagent | *rab9* | This paper | PCR primers | Fwd: 5'- GGCACTATGACGAACA TGCGG; Rev: 5'-tttgcagcactgggaaatccg |
| Sequence-based reagent | *rab10* | This paper | PCR primers | Fwd: 5'- atatctcttgtcacctgcgcc; Rev: 5'-cgaccaccatccatcgttcgg |
| Sequence-based reagent | *rab14* | This paper | PCR primers | Fwd: 5'-gggGCCAG TTCGAGAAAGGG; Rev: 5'-CACGAGCACTGATCC TTGGC |
| Sequence-based reagent | *rab18* | This paper | PCR primers | Fwd: 5'-AAACAAAGCAGCAAGGTGGC; Rev: 5'-CTCCTCGTCGATCTTG TTGCC |
| Sequence-based reagent | *rab19* | This paper | PCR primers | Fwd: 5'- CCAG TTAACGGCCAGAACGG; Rev: 5'-TTGCCTCTCTGAGCA TTGCC |
| Sequence-based reagent | *rab21* | This paper | PCR primers | Fwd: 5'- CAATGGGAACGGC TAAATGCC; Rev: 5'-caacatttaTCGCC-GAGTGCC |
| Sequence-based reagent | *rab23* | This paper | PCR primers | Fwd: 5'- CACCTGCCGGCTTAGA TGCG; Rev: 5'-GAGATA TCGGAACCGGCCCG |
| Sequence-based reagent | *rab26* | This paper | PCR primers | Fwd: 5'- CGATGAAGTGGACA TGCACCC; Rev: 5'-tgcacttgaacttcactggcg |
| Sequence-based reagent | *rab30* | This paper | PCR primers | Fwd: 5'- ACCCAGCGAC TCAAAAACCC; Rev: 5'-GCTGCACAGTTTCCAGA TCCG |
| Sequence-based reagent | *rab32* | This paper | PCR primers | Fwd: 5'-GTAGACACGGGTCATG TTGCC; Rev: 5'-accagcaaatctcagtgcgg |
| Sequence-based reagent | *rab35* | This paper | PCR primers | Fwd: 5'- CGAATCG TAAGCCAAGAACCC; Rev: 5'-ACTAATGGTGACGCAC TGGC |
| Sequence-based reagent | *rab39* | This paper | PCR primers | Fwd: 5'-TAACAACCACCAGCGACAGCC; Rev: 5'-CGTATACCTCGTG TGACTGGC |

*Continued on next page*

*Continued*

| Reagent type (species) or resource | Designation | Source or reference | Identifiers | Additional information |
|---|---|---|---|---|
| Sequence-based reagent | *rab40* | This paper | PCR primers | Fwd: 5'- caatgagtaaacccctagcgg; Rev: 5'-TGGGTATGGGTATGGTATGGG |
| Sequence-based reagent | *rabX1* | This paper | PCR primers | Fwd: 5'- GTGCCCAAGAAATCAGACGC; Rev: 5'-AGTCAGATGGGCTTA-GAGCG |
| Sequence-based reagent | *rabX4* | This paper | PCR primers | Fwd: 5'- CTGTAACCGAAAACCTCCGC; Rev: 5'-CAACTTGCTCAGGTTCTGCG |
| Sequence-based reagent | *rabX6* | This paper | PCR primers | Fwd: 5'- GTCGCACTGTTGTTGTCGCC; Rev: 5'-CTCTGCGTGAGCATTGAGCC |
| Sequence-based reagent | Reverse primer in Gal4-region | This paper | PCR primers | 5'-CGGTGAGTGCACGATAGGGC |
| Sequence-based reagent | Second reverse primer in Gal4-region | This paper | PCR primers | 5'-CAATGGCACAGGTGAAGGCC |
| Sequence-based reagent | Reverse primer in RFP-region | This paper | PCR primers | 5'- GCTGCACAGGCTTCTTTGCC |
| Sequence-based reagent | Second reverse primer in RFP-region | This paper | PCR primers | 5'- ACAATCGCATGCTTGACGGC |
| Sequence-based reagent | Forward primer in RFP-region | This paper | PCR primers | 5'- GGCTCTGAAGCTGAAAGACGG |
| Sequence-based reagent | Forward primer in dsRed-region | This paper | PCR primers | 5'- ATGGTTACAAATAAAGCAATAGCATC |
| Sequence-based reagent | Reverse primer behind right-arm of inserted dsRed-cassette | This paper | PCR primers | 5'-AAACCACAGCCCATAGACG |
| Commercial assay or kit | SapphireAmp Fast PCR Master Mix | Takara Bio Group | Cat. #: RR350A | |
| Commercial assay or kit | Phusion High-Fidelity PCR kit | Thermo Fisher Scientific Inc (Waltham, MA, USA) | Cat. #: F553S | |
| Commercial assay or kit | NucleoSpin Gel and PCR Clean–up | Macherey-Nagel (Düren, Germany) | Cat. #: 740609.50 | Mini kit for gel extraction and PCR clean-up |
| Software, algorithm | ImageJ | National Institutes of Health (NIH) | https://imagej.nih.gov/ij/ | |
| Software, algorithm | Imaris | Bitplane (Zurich, Switzerland) | https://imaris.oxinst.com/packages | |
| Software, algorithm | Amira | Thermo Fisher Scientific Inc (Waltham, MA, USA) | https://www.thermofisher.com/de/de/home/industrial/electron-microscopy/electron-microscopy-instruments-workflow-solutions/3d-visualization-analysis-software.html | |
| Software, algorithm | Adobe Photoshop | Adobe Inc (San Jose, CA, USA) | https://www.adobe.com/products/photoshop.html | |
| Software, algorithm | Adobe Illustrator | Adobe Inc (San Jose, CA, USA) | https://www.adobe.com/products/illustrator.html | |
| Software, algorithm | RStudio | RStudio Inc (Boston, MA, USA) | https://rstudio.com/products/rstudio/ | |
| Software, algorithm | GraphPad Prism | GraphPad Software Inc (San Diego, CA, USA) | https://www.graphpad.com/scientific-software/prism/ | |
| Software, algorithm | AxoScope | Molecular Devices LLC. (San Jose, CA, USA) | https://www.molecular devices.com/ | |

*Continued on next page*

*Continued*

| Reagent type (species) or resource | Designation | Source or reference | Identifiers | Additional information |
| --- | --- | --- | --- | --- |
| Software, algorithm | SnapGene | GSL Biotech LLC (Chicago, IL, USA) | https://www.snapgene.com/ | |
| Other | Toto-3 stain | Thermo Fisher Scientific Inc (Waltham, MA, USA) | Cat. #: T3604 | TOTO-3 Iodide (642/660); IHC (1:1000) |
| Other | Phalloidin stain | Abcam (Cambridge, UK) | Cat. #: ab176752 | Phalloidin-iFluor 405; IHC (1:250) |
| Other | SDS-polyacrylamide Gel | Bio-Rad Laboratories, Inc (Hercules, CA, USA) | Cat. #: 4561083 | 4–15% Mini-PROTEAN TGX Precast Gels |
| Other | PVDF membrane | Bio-Rad Laboratories, Inc (Hercules, CA, USA) | Cat. #: 162–0177 | |
| Other | Clarity Western ECL Substrate | Bio-Rad Laboratories, Inc (Hercules, CA, USA) | Cat. #: 170–5060 | |
| Other | Insect needles | Entomoravia (Slavkov u Brna, Czech Republic) | https://entomoravia.eu/ | Austerlitz insect needles; ø 0.1 mm |

## Fly husbandry and genetics

Flies were raised on molasses formulation food. Stocks were kept at room temperature (22–23°C) in non-crowded conditions, which we defined as 'normal laboratory conditions'. Flies were mostly raised at 25°C or 18°C and 29°C (developmental timing assay).

For the rescue of *rab30* infertility we used: *rab30* $^{Gal4-KI}$, UAS-YFP-Rab30WT.

For the developmental assays, the following deficiency lines were used: *rab3*-Df (Bloomington stock #8909), *rab4*-Df (Bloomington stock #38465), *rab9*-Df (Bloomington stock #7849), *rab10*-Df (Bloomington stock #29995), *rab14*-Df (Bloomington stock #7518), *rab19*-Df (Bloomington stock #7591), *rab32*-Df (Bloomington stock #23664), *rab39*-Df (Bloomington stock #26563), *rab40*-Df (Bloomington stock #26578), *rabX1*-Df (Bloomington stock #26513), *rabX4*-Df (Bloomington stock #25024), and *rabX6*-Df (Bloomington stock #8048). yw was used as wild type control.

For the analysis of the expression pattern of endogenously tagged Rab GTPases in pupae and 1 day-old adults, the following homozygous *Drosophila* lines were used: EYFP-Rab3, EYFP-Rab4, EYFP-Rab9, EYFP-Rab19, EYFP-Rab21, EYFP-Rab23, EYFP-Rab26, EYFP-Rab27, EYFP-Rab32, EYFP-Rab40, EYFP-RabX1, EYFP-RabX4 (EYFP-RabX4/TM6B for adult brain analysis), and EYFP-RabX6 (*Dunst et al., 2015*).

For the analysis of the identity of the Chaoptin-positive accumulations in *rab26* lamina after 4 days of light stimulation, following *Drosophila* lines were used: *rab26* and yw as wild type control. For the rescue of the Chaoptin-accumulation phenotype, following *Drosophila* lines were used: ;UAS-YFP-Rab26WT/+; *rab26*, lGMR-Gal4, UAS-white RNAi/*rab26* as well as ;;lGMR-Gal4, UAS-white RNAi and ;;*rab26*, lGMR-Gal4, UAS-white RNAi/*rab26* as negative and positive control, respectively. To test the efficiency of the Rab26 RNAi line KK107584 (VDRC stock ID: 101330) the following fly lines were used: UAS-Rab26 RNAi/+; elav-Gal4/+ and UAS-YFP-Rab26WT/UAS-Rab26 RNAi; lGMR-Gal4, UAS-white RNAi/+. To reproduce the *rab26* mutant phenotype, the following *Drosophila* line was used: UAS-Rab26 RNAi/+; lGMR-Gal4, UAS-white RNAi. For the analysis of possible colocalization between Rab26-positive compartments and synaptic vesicle markers as well as endomembrane trafficking markers, following *Drosophila* lines were used: ;elav-Gal4/UAS-YFP-Rab26WT;, ;elav-Gal4/UAS-YFP-Rab26CA;, ;elav-Gal4/UAS-YFP-Rab26DN;, ;sGMR-Gal4/UAS-YFP-Rab26WT; and ;sGMR-Gal4/UAS-YFP-Rab26CA;. For the comparison of the anti-Rab26 antibody labeling with the YFP-knock in line, the following *Drosophila* line was used: ;UAS-YFP-Rab26WT/+;rab26$^{exon1}$-Gal4/+. For the Rab26 lamina localization analysis, the 31C06-Gal4 (L4-Gal4) as well as Split-Gal4 Lawf1 (SS00772) and Lawf2 (SS00698) lines were crossed to ;UAS-CD4-tdGFP;. For the photoreceptor-specific knock down of Dα4 receptor subunit the following fly line was used: ;UAS-rye RNAi; UAS-Dicer2/lGMR-Gal4,UAS-white-RNAi. The ;UAS-rye RNAi; UAS-Dicer2 stock was a gift from the Amita Sehgal lab.

## Generation of null mutant flies

All CRISPR/Cas9-mediated *rab* mutants, except *rab18* and *rab26*, were generated by WellGenetics Inc (Taipei, Taiwan), by homology-dependent repair (HDR) using two guide RNAs and a dsDNA plasmid donor (*Kondo and Ueda, 2013*). Briefly, upstream and downstream gRNA sequences were cloned into a U6 promoter plasmid. For repair, a cassette, containing two loxP-sites flanking a 3xP3-RFP with two homology arms was cloned into a donor template (pUC57-Kan). A control strain (w[1118]) was injected with the donor template as well as specific *rab*-targeting gRNAs and hs-Cas9. F1 progeny positive for the positive selection marker, 3xP3-RFP, were further validated by genomic PCR and sequencing. The CRISPR null mutants were validated as described in the next section. gRNA sequences as well as specifics on the different CRISPR mutants are as follows:

- **rab9**: Replacement of 2446 bp region, +98 bp relative to ATG to +111 bp relative to the first bp of *rab9* stop codon, by floxable cassette. Upstream gRNA sequence: GTTGTTCTCCTCG TAGCGAT, downstream gRNA sequence: ATTCCAGTCCGCGGAGGGGC.
- **rab10**: Replacement of 1644 bp region, +57 bp relative to ATG to +70 bp relative to the fist bp of *rab10* stop codon, by cassette, which contains three stop codons upstream of floxable 3xP3-RFP. Upstream gRNA sequence: CTGATCGGTGATTCAGGAGT, downstream gRNA sequence: GAACGGGGCGTGGTTTGGCC.
- **rab14**: Replacement of 930 bp region, −17 bp relative to ATG of *rab14-RB* isoform to −61 bp relative to the first bp of *rab14* stop codon, by floxable cassette. Upstream gRNA sequence: GATGAGCAAAGTGCGCAGCG, downstream gRNA sequence: GAAG TTCGCGACGGCTGCGA.
- **rab21**: Replacement of 608 bp region, +12 bp relative to ATG of *rab21-RD* isoform to −109 bp relative to first bp of *rab21* stop codon, by floxable cassette. Upstream gRNA sequence: CAATGAGCTCGAGCAGAACG, downstream gRNA sequence: GACTCGCA TCCGGTTGCCGT.
- **rab23**: Replacement of 1700 bp region, −35 bp relative to ATG to +173 bp relative to the first bp of *rab23* stop codon, by floxable cassette. Upstream gRNA sequence: CAATCAAACACC TGGGCGAG, downstream gRNA sequence: CATGTCTGAACCACATCACG.
- **rab35**: Replacement of 816 bp region, −24 bp relative to ATG of *rab35-RC* isoform to +20 bp relative to the first bp of *rab35* stop codon, by floxable cassette. Upstream gRNA sequence: CAGCAATGTCATATGCCGAA, downstream gRNA sequence: AGGTGAAAGCGGC TCCGGCA.
- **rab39**: Replacement of 898 bp region, +92 bp relative to ATG to −93 bp relative to the first bp of *rab39* stop codon, by floxable cassette. Upstream gRNA sequence: CACAGACGGCAAATTCGCCG, downstream gRNA sequence: TCGATCCGGCGAA TATAAGG.
- **rab40**: Replacement of 1407 bp region, +2 bp relative to ATG to −93 bp to the first bp of *rab40* stop codon, by floxable cassette. Upstream gRNA sequence: CCTTGGTCATGG TTCCCATG, downstream gRNA sequence: TTGAGCGTCGACTTCACCGA.
- **rabX4**: Replacement of 962 bp, −2 bp relative to ATG to −61 bp to first bp of *rabx4* stop codon, by floxable cassette. This results in the deletion of the entire coding sequence. Upstream gRNA sequence: CTCCGCCAGCTCCGTCAACA, downstream gRNA sequence: AAGAAATCACCCGGCTCCAA.
- **rab18**: For the generation of the *rab18* null mutant, first a *rab18* sgRNA-expressing plasmid (pBFv-U6.2-rab18-sgRNA) was generated. For this, *rab18* sgRNA sequence 5'-GGTGA TCGGGGAAAGCGGCG (directly after the *rab18* start codon) was cloned into BbsI-digested pBFv-U6.2 plasmid. Second, a pCR8-rab18-3xP3-RFP plasmid was generated by soeing PCR and restriction enzyme digestion. For this, two 500 bp homology arms (HA) around the *rab18* sgRNA targeting site were amplified, using the following primers: left HA fwd: TCC TAAATTTATGATATTTTATAATTATTT; left HA rev: CTGGACTTGCCTCGAGTTTTTTAGATCTG TGTGGTTTGAGCTCCGCTT; right HA fwd: CAAACCACACAGATCTAAAAAACTCGAGG-CAAGTCCAGGTGCAGTCCC; right HA rev: CGAACTGATCGCATTTGGCT. The resulting PCR product was then cloned into pCR8 vector (pCR8-rab18LA+RA). The 3xP3-RFP cassette, containing three stop codons upstream of the RFP, was cloned into pCR8-rab18LA+RA by *Bgl*II and *Xho*I double digestion to get the final pCR8-rab18-3xP3RFP plasmid. *Nanos-Cas9* fly embryos were co-injected with the two plasmids pBFv-U6.2-rab18-sgRNA and pCR8-rab18-3xP3RFP. F1 progeny positive for the selection marker, 3xP3-RFP, were further validated by genomic PCR.

- **rab26**: Replacement of 9760 bp region, - 125 bp relative to ATG to +1310 bp to the end of coding exon 2, by positive selection marker 3xP3-dsRed flanked by loxP-sites. This leads to the complete deletion of ATG1 (exon 1) and ATG2 (exon 2) of *rab26* gene. Briefly, a *rab26* sgRNA-expressing plasmid was generated by cloning the *rab26* sgRNA 5'-GACAG TTTCGGAGTTAATTA into a BbsI-digested U6-BbsI-chiRNA plasmid (Addgene, plasmid #45946, donated by Kate O'Connor-Giles lab). *Nanos-Cas9* fly embryos were co-injected with the *rab26* sgRNA containing U6-chiRNA plasmid and the pHD-DsRed-attP plasmid (donated by Kate O'Connor-Giles lab). F1 progeny positive for the selection marker, 3xP3-dsRed, were further validated by genomic PCR.

In addition, six *rab* mutants (*rab2*, *rab4*, *rab19*, *rab30*, *rabX1*, and *rabX6*) were generated by ends-out homologous recombination based on previously generated Gal4 knock-ins in large genomic fragments (*Chan et al., 2011*). All *rab* mutants generated by ends-out homologous recombination are 'ORF knock-ins' (replacing the entire open reading frame), except for *rab4*, which is an 'ATG knock-in' (replacing the first exon including the start codon). The methods used for the replacements in the endogenous loci have been described previously in detail (*Chan et al., 2012*; *Chan et al., 2011*).

## Verification of *rab* null mutants by PCR

The newly generated *rab* null mutants were confirmed by genomic PCR, either using Phusion High-Fidelity PCR Kit (Thermo Fisher Scientific) (majority of *rab* mutants) or the SapphireAmp Fast PCR Master Mix (TaKaRa) (*rab26*). The following primer pairs, flanking the gene or inserted cassette, were used for the validation: *rab2* (Fwd: 5'-TGGCCACACTGTCGCTAGCC and Rev: 5'-CGCCTCCTC TACGTTGGCAG), *rab4* (Fwd: 5'- GGTTTTGATCGTGTCCTGCG and Rev: 5'-AGACAACTC TTACCGCTGCC), *rab9* (Fwd: 5'- GGCACTATGACGAACATGCGG and Rev: 5'-TTTGCAGCAC TGGGAAATCCG), *rab10* (Fwd: 5'- ATATCTCTTGTCACCTGCGCC and Rev: 5'-CGACCACCATCCA TCGTTCGG), *rab14* (Fwd: 5'-gggGCCAGTTCGAGAAAGGG and Rev: 5'-CACGAGCACTGATCC TTGGC), *rab18* (Fwd: 5'- AAACAAAGCAGCAAGGTGGC and Rev: 5'-CTCCTCGTCGATCTTG TTGCC), *rab19* (Fwd: 5'- CCAGTTAACGGCCAGAACGG and Rev: 5'-TTGCCTCTCTGAGCATTGCC ), *rab21* (Fwd: 5'- CAATGGGAACGGCTAAATGCC and Rev: 5'-CAACATTTATCGCCGAGTGCC), *rab23* (Fwd: 5'- CACCTGCCGGCTTAGATGCG and Rev: 5'-GAGATATCGGAACCGGCCCG), *rab26* (Fwd: 5'- CGATGAAGTGGACATGCACCC and Rev: 5'-TGCACTTGAACTTCACTGGCG), *rab30* (Fwd: 5'- ACCCAGCGACTCAAAAACCC and Rev: 5'-GCTGCACAGTTTCCAGATCCG), *rab35* (Fwd: 5'- CGAATCGTAAGCCAAGAACCC and Rev: 5'-ACTAATGGTGACGCACTGGC), *rab39* (Fwd: 5'-TAACAACCACCAGCGACAGCC and Rev: 5'-CGTATACCTCGTGTGACTGGC), *rab40* (Fwd: 5'- caat-gagtaaacccctagcgg and Rev: 5'-TGGGTATGGGTATGGTATGGG), *rabX1* (Fwd: 5'- GTGCCCAA-GAAATCAGACGC and Rev: 5'-AGTCAGATGGGCTTAGAGCG), *rabX4* (Fwd: 5'- CTG TAACCGAAAACCTCCGC and Rev: 5'-CAACTTGCTCAGGTTCTGCG), and *rabX6* (Fwd: 5'- G TCGCACTGTTGTTGTCGCC and Rev: 5'-CTCTGCGTGAGCATTGAGCC). For the validation of the mutants generated by homologous recombination, the following cassette-specific primers were used: Reverse primer in Gal4-region: 5'-CGGTGAGTGCACGATAGGGC (*rab2*, *rab4*, *rabX1*), second reverse primer in Gal4-region: 5'-CAATGGCACAGGTGAAGGCC (*rab19*, *rab30*, *rabX6*). The following cassette specific primers were used for the validation of CRISPR-generated null mutants: Reverse primer in RFP-region: 5'- GCTGCACAGGCTTCTTTGCC (*rab9*, *rab10*, *rab14*, *rab18*, *rab39*, *rabX4*), second reverse primer in RFP-region: 5'- ACAATCGCATGCTTGACGGC (*rab21*, *rab35*, *rab40*), forward primer in RFP-region: 5'- GGCTCTGAAGCTGAAAGACGG (*rab23*), forward primer in dsRed-region: 5'- ATGGTTACAAATAAAGCAATAGCATC (*rab26*) and reverse primer behind right-arm of inserted dsRed-cassette: 5'-AAACCACAGCCCATAGACG (*rab26*). The CRISPR null mutants were independently validated in our lab and by WellGenetics Inc (Taipei, Taiwan). All primers were designed with SnapGene (GSL Biotech LLC).

## Immunohistochemistry

Pupal and adult eye-brain complexes were dissected and collected in ice-cold PBS. The tissues were fixed in PBS with 4% paraformaldehyde for 30 min and washed in PBST (PBS + 1% Triton X-100). Wandering L3 larvae were immobilized at their abdomen and mouth hooks on a Sylgard-filled dissection dish, using insect needles (ø0.1 mm, Austerlitz insect pins). Larvae were dissected, from the dorsal side, in ice-cold 1x Schneider's *Drosophila* Medium (Thermo Fisher Scientific) and immediately

fixed in PBS with 4% paraformaldehyde for 10 min. After fixation, the gut and main trachea were carefully removed and the larval filets washed in PBS-Tween (PBS + 0.1% Tween).

The following primary antibodies were used: rabbit anti-Rab5 (1:1000, Abcam), rabbit anti-Rab7 (1:1000, gift from P. Dolph), mouse anti-Rab11 (1:500, BD Transduction Laboratories), guinea pig anti-Rab26 (1:2000 (IHC), 1:1000 (WB), made for this study), mouse anti-Syt1 (1:1000, DSHB), rabbit anti GABARAP+GABARAPL1+GABARAPL2 (Atg8) (1:100, Abcam), rabbit anti-Syx7/Avalanche (1:1000, gift from H. Krämer), guinea pig anti-Hrs (1:300, gift from H. Bellen), rabbit anti-HRP (1:500, Jackson ImmunoResearch Laboratories), rabbit anti-DPAK (1:2000), rat anti-Dα7 (1:2000, gift from H. Bellen), rat anti-nCadherin (1:100, DSHB), guinea pig anti-V100 (1:1000, *Hiesinger et al., 2005*), mouse anti-CSP (1:50, DSHB), mouse anti-ChAT (1:100, DSHB), mouse anti-nc82 (1:20, DSHB), rabbit anti-ebony (1:200), mouse anti-Chaoptin (24B10) (1:50, DSHB) and rabbit anti-DCP-1 (1:100; Cell Signaling Technology). Secondary antibodies used were Donkey anti-mouse DyLight 405, Goat anti-mouse Alexa 488, Goat anti-guinea pig Alexa 488, Goat anti-rat Alexa 488, Goat anti-rabbit Cy3, Goat anti-rabbit Alexa 647, Goat anti-rat Alexa 647, Goat anti-mouse Cy5, Goat anti-rat Cy5 (1:500; Jackson ImmunoResearch Laboratories), Goat anti-guinea pig HRP-linked (1:5000, Jackson ImmunoResearch Laboratories), Goat anti-guinea pig Cy5 (1:500, Abcam), Phalloidin-iFluor 405 (1:250; Abcam) and Toto-3 (1:1000; Thermo Fisher Scientific).

All samples were mounted in Vectashield mounting medium (Vector Laboratories). Larval filet preparations were incubated in Vectashield for at least 30 min at 4°C prior to mounting. To fully expose lamina photoreceptor terminals, pupal brains were mounted with their dorsal side up.

## Generation of rab26 antibody

The cDNA sequence corresponding to amino acids 1–192 of *rab26* was amplified by PCR and cloned into the pET28a (Invitrogen) vector for protein expression. Guinea pig antibodies against this domain were raised by Cocalico Biomedicals, Inc using the purified recombinant protein.

## Confocal microscopy, image processing, and quantification

All microscopy was performed using a Leica TCS SP8 X (white laser) with 20x and 63x Glycerol objectives (NA = 1.3). Leica image files were visualized and processed using Imaris (Bitplane) and Amira (Thermo Fisher Scientific). Postprocessing was performed using ImageJ (National Institute of Health), and Photoshop (CS6, Adobe Inc). Data was plotted using Illustrator (CS6, Adobe Inc), Photoshop (CS6, Adobe Inc) and GraphPad Prism 8.3.0 (GraphPad Software Inc).

For Chaoptin-accumulation and rhabdomere morphology experiments, all quantification was performed manually on single slices and only individually discernible compartments or rhabdomeres were counted. Only Chaoptin-accumulations in the central region of the lamina (length 115 μm and depth 27 μm) were quantified. For the rhabdomere analysis, the measurement tool from ImageJ was used. For Rhabdomere quantifications, 150 outer rhabdomeres were analyzed the following way: The longest (a) as well as the shortest (b) rhabdomere diameter was measured using the ImageJ measurement tool. For the shape analysis, the longest diameter was divided by the shortest (shape = a/b). For the area analysis, the following mathematical formula was used: $area = pi * \left( \frac{a}{2} \right) * \left( \frac{b}{2} \right)$.

The rhabdomere area ratio was calculated by dividing the area of newly hatched flies by the area of flies after 2 days of light stimulation. Area is more variable than shape in wild type and only significant changes outside the standard deviation range of the wild type control were scored. The statistical analyses were performed using RStudio (RStudio Inc) and GraphPad Prism 8.3.0 (GraphPad Software Inc), and the specific statistical tests used as well as sample numbers for experiments are indicated in the respective figure legends.

## Biochemistry

Proteins were extracted from 20 adult fly brains per genotype in RIPA buffer containing 150 mM NaCl, 0.1% Triton X–100 (Sigma), 0.1% SDS (Amresco), 50 mM Tris-HCL and 1x complete protease inhibitors (Sigma), pH 8. Samples were incubated on ice for 20 min and centrifuged at 16,000 RCF, 10 min at 4°C to remove cell debris. Laemmli buffer (Bio-Rad Laboratories) was added to the supernatant. After incubation for 5 min at 95°C, the samples were loaded on a 4–15% SDS-polyacrylamide gel (Bio Rad Laboratories) and then transferred to PVDF membrane (Bio-Rad Laboratories). Primary

antibody used was guinea pig anti-Rab26 (1:1000) and corresponding secondary was used 1:5000 (Abcam). The signals were detected with Clarity Western ECL (Bio-Rad Laboratories).

## Backcrossing of *rab* mutant flies

Serial backcrossing to a wild type (yw) background was performed for three consecutive generations. The single *rab* mutants as well as the respective balancer chromosomes, used to generate the final stocks, were backcrossed to the same genetic background. All mutant alleles, except *rab3* and *rab32*, could be traced by their red fluorescent marker. Where direct tracing was not possible, backcrossing was performed 'blindly' and after three generations roughly 100 separate single (fe-)male stocks were generated and subsequently sequenced to identify the backcrossed *rab3* and *rab32* mutants.

The genomic DNA was amplified using the Phusion High-Fidelity PCR Kit (Thermo Fisher Scientific) with the following primers for *rab3* (Fwd: 5'-ACACTGAGGCGAGCTTACGC and Rev: 5'- CTAC TACCGAGGAGCGATGGG) and *rab32* (Fwd: 5'-GTAGACACGGGTCATGTTGCC and Rev: 5'-accag-caaatctcagtgcgg). The amplified DNA was extracted from agarose gel, cleaned using the Nucleo-Spin Gel and PCR Clean-up kit (Macherey-Nagel) and send for sequencing to Microsynth Seqlab GmbH (Göttingen, Germany). Sequencing results were visualized using SnapGene (GSL Biotech LLC). All primers were designed with SnapGene (GSL Biotech LLC).

## Developmental assays

For the analysis of developmental timing of homozygous, viable *rab* mutants, three crosses with equal number of flies (ratio female to male ~2:1) and same genotype were set up a few days prior to the start of the experiment, to ensure good egg laying. Of each of those, again three equal groups were formed and egg laying was allowed for 24 hr at room temperature. Egg containing vials were then shifted to the respective temperatures (18℃, 25℃, or 29℃), while the parental flies remained at room temperature for the duration of the experiment. The shifting of egg containing vials was repeated six more times, leading to a total of 21 'experimental' vials per temperature per genotype. Developing flies were kept at the respective temperatures until three days after they hatched, and the total number of hatched offspring was counted.

To study the effect of temperature stress on fly wing development, *rab* null mutants were reared at 18℃ and 29℃. All mutant lines were set up with 10 females and three males and kept in their vials for 48 hr of egg laying, so as to prevent overcrowding in the vials. Adult female flies were collected not earlier than 24 hr after eclosion and placed in a 1:1 solution of glycerol:ethanol for a minimum of several hours, after which the wings were removed at the joint and mounted in the same solution. Wings were imaged with a Zeiss Cell Observer microscope and their size measured in ImageJ (National Institute of Health).

To validate the phenotypes of the developmental assay, backcrossed mutants as well as transheterozygotes of mutant chromosomes over deficiency chromosomes were used. All deficiency chromosomes were placed over a fluorescent balancer prior to the assay, to allow for identification. We did not succeed in identifying and validating a deficiency line for *rab18*. Homozygous backcrossed *rab32* females are lethal at 29℃, therefore no wing surface area measurements are available. All conditions, like temperature and number of parental flies, were kept same.

## Neuronal stimulation with white light and electroretinogram (ERG) recordings

Newly eclosed adults were either placed in a box for constant white light stimulation or placed in light-sealed vials (in the same box) for constant darkness. The lightbox contains two opposing high-intensity warm white light LED-stripe panels, each emitting ~1600 lumen (beam angle = 120°, distance between light source and vials = 16 cm). Temperature (22℃) and humidity (59%) inside the box were kept constant. Flies were kept inside the box for up to 7 days (wild type sensitization curve) or for 2 and 4 days (function and maintenance experiments).

For the ERG recordings, the flies were anesthetized and reversibly glued on microscope slides using non-toxic school glue. The recording and reference electrodes were filled with 2 M NaCl and placed on the retina and inside the thorax. Flies were exposed to a series of 1 s light/dark pulses provided by a computer-controlled white light-emitting diode system (MC1500; Schott) as previously

reported (*Cherry et al., 2013*). Two different light stimulus intensities, dim ($5.29e^{13}$ photons/cm$^2$/s) and bright ($1.31e^{16}$ photons/cm$^2$/s), were used. Retinal responses were amplified by a Digidata 1440A, filtered through a Warner IE-210, and recorded using AxoScope 10.6 by Molecular Devices. All ERG recordings were performed in non-pigmented, white-eyed flies, which are more sensitive to light stimulation than pigmented ones. A total of 25–30 flies were examined for each genotype, condition, and time point.

For the quantification of the ERG data AxoScope 10.6 by Molecular Devices was used. First, the 'on' transient was quantified, by measuring the difference between the averaged baseline, prior to the onset of the light stimulus, and the peak value of the 'on' transient itself. Second, the depolarization was quantified, by measuring the difference between the baseline prior to stimulation and the depolarization when the signal has reached its plateau in the second half of the 1 s light stimulus prior to the end of the stimulus and repolarization.

### Neuronal stimulation with blue light

Newly eclosed *rdgC*$^{306}$ mutant flies (Bloomington stock #3601) were placed in an illuminated aluminum tube for constant, high-intensity, pure blue light stimulation. The aluminum tube has an outer diameter of 45 mm and a wall thickness of 2.5 mm. It contains one high-intensity blue light LED-stripe, covering the complete inside of the tube and emitting 155 lumen (beam angle = 120°, distance between light source and vials = ~1 cm). Temperature (22°C) and humidity (59%) inside the tube were kept constant. Flies were kept under constant blue light stimulation for 4 consecutive days and afterwards immediately placed in the dark for 2 days.

## Acknowledgements

We would like to thank members of the Boutros, Hiesinger, Wernet and Hassan labs for their support and helpful discussions. We thank Hugo Bellen, Helmut Krämer, Amita Sehgal and the Developmental Hybridoma Bank for flies and reagents. This work was supported by grants from the NIH (RO1EY018884), the German Research Foundation (DFG, SFB/TRR186) to PRH and the German Research Foundation (DFG, SFB/TRR186) to MB.

## Additional information

### Competing interests

P Robin Hiesinger: Reviewing editor, *eLife*. The other authors declare that no competing interests exist.

### Funding

| Funder | Grant reference number | Author |
| --- | --- | --- |
| National Institutes of Health | RO1EY018884 | P Robin Hiesinger |
| Deutsche Forschungsgemeinschaft | SFB/TRR186 | P Robin Hiesinger |
| Deutsche Forschungsgemeinschaft | SFB/TRR186 | Michael Boutros |

The funders had no role in study design, data collection and interpretation, or the decision to submit the work for publication.

### Author contributions

Friederike E Kohrs, Ilsa-Maria Daumann, Bojana Pavlovic, Filip Port, Formal analysis, Investigation, Methodology, Writing - original draft, Writing - review and editing; Eugene Jennifer Jin, Conceptualization, Formal analysis, Investigation, Methodology; F Ridvan Kiral, Formal analysis, Investigation, Methodology; Shih-Ching Lin, Heike Wolfenberg, Thomas F Mathejczyk, Methodology; Gerit A Linneweber, Formal analysis, Validation, Methodology; Gerit Linneweber was added as an author during the revision with the approval of all authors regarding inclusion and position in the author list,

because he substantially helped with the backcrossing required for the revision and all data analyses associated with the new developmental data in Figure 2 plus supplements; Chih-Chiang Chan, Conceptualization, Supervision, Funding acquisition, Investigation, Methodology, Writing - original draft, Writing - review and editing; Michael Boutros, P Robin Hiesinger, Conceptualization, Formal analysis, Supervision, Funding acquisition, Writing - original draft, Project administration, Writing - review and editing

### Author ORCIDs
Shih-Ching Lin 🔟 http://orcid.org/0000-0003-2960-5348
Filip Port 🔟 http://orcid.org/0000-0002-5157-4835
Chih-Chiang Chan 🔟 http://orcid.org/0000-0003-2626-3805
Michael Boutros 🔟 http://orcid.org/0000-0002-9458-817X
P Robin Hiesinger 🔟 https://orcid.org/0000-0003-4698-3527

### Decision letter and Author response
Decision letter https://doi.org/10.7554/eLife.59594.sa1
Author response https://doi.org/10.7554/eLife.59594.sa2

---

# Additional files

### Supplementary files
• Supplementary file 1. Notes on pupal and adult expression patterns of nervous system-enriched Rabs based on endogenously tagged Rabs generated by *Dunst et al., 2015*. (A) Expression notes on optic lobe expression at 40% pupal development. (B) Expression notes on adult brains. The expression patterns are shown in *Figure 1—figure supplements 2–3*.

• Supplementary file 2. Tissue localization of Rab proteins in humans, rodents (*mus musculus*, *rattus norvegicus*, white New Zealand rabbits (*Oryctolagus cuniculus*)) and *Drosophila melanogaster* based on RNA- and protein-level expression. For the human protein atlas (www.proteinatlas.org based on Fagerberg et al., 2014) 27 tissues were analyzed. The data was summarized in the following way: "ubiquitous" (detected in all tissue/region/cell types), "widespread" (detected in at least a third but not all tissue/region/cell types), "restricted" (detected in more than one but less than one third of tissue/region/cell types). The classifications "tissue specific", "tissue enriched", "group enriched" and "uncertain" were used as described in the human protein atlas. Regarding the data of the mouse embryo (E 14.5) transcriptome atlas (www.eurexpress.org based on Diez-Roux et al., 2011) the original classifications were adopted: "regional signal" (signal detected in a limited number of discrete locations), "no regional signal" (in all tissues or not detectable) or "not detected". Out of the analyzed tissues "brain, spinal cord, CNS nerves, peripheral nervous system, ganglia" were grouped as nervous system and "gut, stomach, liver, pancreas" as intestines. For the flyatlas2 (www.flyatlas.gla.ac.uk, see also based on Leader et al., 2018) only data of female adults were considered. "Head, brain and thoracicoabdominal ganglion" were grouped as "nervous system high". The following abbreviations were used: human (H), rodent (R), *Drosophila melanogaster* (Dm), embryo (E), larva (L), adult (A), *Mus musculus* (Mm), *Rattus norvegicus* (Rn), *Oryctolagus cuniculus* (Oc), *cell culture* (CC). Asterisks indicate if the Rab is specific to Hominidae (*), specific to primates (**) or specific to primates and dolphins (***).

• Supplementary file 3. Function, subcellular localization, and mutant viability of Rab GTPases in mammals, *Saccharomyces cerevisiae* and *Drosophila melanogaster*. Mouse knockout models were listed for the mammalian *rab* GTPase mutants. Among primary publications, the International Mouse Phenotype Consortium (https://www.mousephenotype.org/) was used for information on the viability of mouse knockout models Information on *Drosophila* mutant viability is based on this study, if not stated otherwise in the table. Only viability / lethality for homozygous mutants was listed. The following abbreviations were used: *Drosophila melanogaster* (Dm), endoplasmic reticulum (ER), glucose transporter type 4 (GLUT4), insulin-producing cells (IPCs), Jun-N-terminal kinase (JNK), knockout (KO), mammals (M), matrix metalloproteinases (MMP), multivesicular bodies(MVBs), neuromuscular junction (NMJ), planar cell polarity (PCP), plasma membrane (PM), *Saccharomyces cerevisiae*(Sc),

trans-Golgi network (TGN), 37tyrosinase-related protein-1 (Tyrp-1), ventral nerve cord (VNC). Asterisks indicate if the Rab isspecific to Hominidae (*), specific to primates (**) or specific to primates and dolphins (***).

• Supplementary file 4. Quantitative analysis of the developmental timing assay at different temperatures. (A) Summary of developmental time for wild type and all fertile, homozygous viable *rab* mutants at 18℃, 25℃, and 29℃. Listed are number of days (after 24 hr of egg collection) until first 1st instar larvae, pupae, or adults appear, as well as total number of adults hatched and number of adults per vial. Days are given in mean ± SEM. (B) Summary of developmental time for wild type and tested backcrossed *rab* mutants at 18℃, 25℃, and 29℃. Listed are number of days (after 24 hr of egg collection) until first 1st instar larvae, pupae, or adults appear, as well as total number of adults hatched and number of adults per vial. Days are given in mean ± SEM. (C) Summary of developmental time for wild type and tested *rab* mutants over deficiencies at 18℃, 25℃ and 29℃. Listed are number of days (after 24 hr of egg collection) until first 1st instar larvae, pupae, or adults appear, as well as total number of adults hatched and number of adults per vial. Days are given in mean ± SEM.

• Transparent reporting form

### Data availability

All data generated or analysed during this study are included in the manuscript and supporting files.

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
