## [Decision Letter]

**Acceptance summary:**

This work reports the generation and characterization of molecularly defined null mutants for all 26 rab genes in *Drosophila*. Loss of 13 nervous system-enriched Rabs yielded viable and fertile flies without obvious morphological defects. However, all 13 mutants differentially affected development when challenged with different temperatures, or neuronal function when challenged with continuous stimulation. The work shows a synaptic maintenance defect following continuous stimulation for six mutants, including an autophagy-independent role of rab26. This is a highly valuable resource for scientists interested in Rab function.

**Decision letter after peer review:**

Thank you for submitting your article "Systematic functional analysis of Rab GTPases reveals limits of neuronal robustness in *Drosophila*" for consideration by *eLife*. Your article has been reviewed by three peer reviewers, including Mani Ramaswami as the Reviewing Editor and Reviewer #1, and the evaluation has been overseen by Utpal Banerjee as the Senior Editor. The following individual involved in review of your submission has agreed to reveal their identity: Suzanne R Pfeffer (Reviewer #3).

The reviewers have discussed the reviews with one another and the Reviewing Editor has drafted this decision to help you prepare a revised submission.

Summary:

Kohrs et al. generated a collection of 26 Rab knockouts in *Drosophila*, to complement their previous systematic Rab expression pattern and localization studies (Chan 2011, PMID22000105; Jin 2012 PMID 22844416, Dunst 2015, PMID 25942626). They make the interesting observation that flies with null mutations in nervous system-enriched Rabs are viable, while null mutants for ubiquitously expressed Rab mutants are lethal. In the first part the paper, they elucidate developmental and broad functional roles of Rabs enriched in the nervous system and, interestingly, identify conditions under which viable rab mutants show strong phenotypes. Together the comprehensive collection of null mutations as well their characterization represent a resource useful and important for *Drosophila* biologists interested in membrane traffic in general, not on their own, but as a key complement to the existing Rab mutants and RNAi tools, YFP-Rab+YFP RNAi or degron tag collection, and the UAS-CA/DN collection. In contrast, the other "resource" section of the paper describing a RUSH Rab toolkit for studying trafficking of Rabs, which is generated through considerable effort, leads to the clear conclusion that substantial further work is needed before this tools can be gainfully utilized. Finally, through more careful analysis of the Rab26 mutant they provide evidence consistent with Rab26 regulating receptor turnover at cholinergic synapses in the visual system of adult flies.

The scientific advances in this paper (viability and wing size of the mutants at different temperatures, ERG recordings at different ages and light exposures) are largely solid descriptive, despite the more in-depth but still not comprehensive characterization of the Rab26 function. However, with appropriate revisions and additions, it may be acceptable as a "tools and resources" paper.

Essential revisions.

1) The detailed characterization of RUSH system reveals several concerns and caveats with respect to its use that make aspect of the work is too preliminary and untested to be published as a Resource in *eLife*. This entire section should therefore be removed.

The authors generated an UAS collection of SBP tagged Rabs to study the trafficking of Rabs in neurons via the RUSH system, which enables biotin-dependent release of Rabs from a sequestered location. The biggest concern is that the biotin-free media required to set up the experiment compromises animal health. In addition to this issue, sequestration of the overexpressed Rabs may deplete Rab effectors from their normal locations. Therefore the experiments start from a non-inert condition where there may be significant background phenotypes or developmental compensation, compromising the interpretation of results. Thus, despite the interesting observation of biotin dependent redistribution in one or two cases, the work makes it clear that careful additional experiments will be essential for each RUSH lines, before they can be used to conditionally control respective Rab activity in vivo. At very minimum one would need to know if the RUSH lines rescue corresponding null mutants.

There are also several specific queries and concerns, we mention in case these are useful to the authors to take this forward.

a) The system is not designed in *Drosophila* to ensure 1:1 expression of the reporter (SBP) and the hook (streptavidin). The UAS constructs are in different chromosomes unlike the bicistronic design used in cultured cells in the original RUSH paper (Boncompain et al., 2012). To address this issue one needs to quantify the level of expression of the reporter and the hook to ensure that the reporter is not expressed at higher levels, leading to unbound reporter in the absence of biotin. One way of doing this is to perform qPCR to measure the abundance of the reporter and the hook.

b) A positive control to show that the RUSH system works properly in *Drosophila*. For example, a good positive control would be to have an UAS myristoylated SBP. In the absence of biotin, this construct should be restricted to the Golgi. After induction with biotin, the majority should be at the plasma membrane. Another positive control would be to label the SBP-tagged and the endogenous cognate Rab. Before induction, all SBP tagged Rab should not co-localize with the endogenous Rab and vice versa. There is a commercially available Rab7 antibody in DSHB that works that could be used for this proof-of-concept experiment.

c) Another concern related to the RUSH system is that significant changes to the Golgi (hook) are observed., while this compartment appears to be stable before and after induction in the original study (Boncompain et al., 2012). This could be a cell specific phenomenon therefore the authors should ensure that in the wildtype the structure of the Golgi is highly dynamic in these cells as well. They could address this by labeling a Golgi resident protein and perform a similar time-lapse image analysis as reported in the manuscript.

d) Is the construct YFP-Rab-SBP (SBP added at the end of the Rab hypervariable domain as indicated in Figure 5?) or Rab-SBP-YFP as indicated in Supplementary figure 6? and when it is released from the Golgi HOOK how does it get prenylated? Is it able to rescue a phenotype? Can it act if not prenylated? Rabs need to associate with membranes to exert their functions and Rab hypervariable domains contribute to effector binding and Rab localization. Supplementary figure 6 needs compartment labeling to show that a released Rab relocalizes to the correct compartment; release from an aggregate is not sufficient (or useful) if the protein is subsequently non-functional.

2) The sections describing the toolkit of molecularly defined null mutants for all 26 rab genes in *Drosophila* and their characterization are clear and valuable, but also require several additional clarifications and controls before publication.

a) The authors test the effects of different temperatures on the development of *Drosophila* in mutants of Rabs enriched in the nervous system. Because mutations are generated via different methods, the genetic backgrounds of those flies should be equalized across all the lines studied (ideally via backcrossing or at least transheterozygotes of independently derived alleles) to exclude unknown variables. The Materials and methods and Results sections do not make it clear if such backcrossing was performed (though the ERG sections indicates that all recordings were performed everything in a w- background). This should be clarified in the manuscript.

b) The authors measure ERG "on" transients to determine if Rab mutants disrupt synaptic transmission. This is a very important experiment, but the dataset for 2 days light appears highly variable. This is an issue, because several Rabs with "not significant" differences seem to have much higher variances, and therefore there may actually be something going on. This high degree of variability is not observed in controls at 2 days light, or in the 0 days or 4 days dark datasets. Could high levels of variability be a result of neuronal death in Rab mutants exposed to 2 days of light? The authors should explore this issue as a source of variability in their dataset by performing something like a TUNEL assay or EM in these Rab mutants, or at least discuss it.

c) Chaoptin staining is used to assess structural differences in the photoreceptor projections in Rab mutants. The representative images used, which are described in the text as having "no phenotype," appear to have decreased Chaoptin staining (e.g. Figure 4A R1-R6 middle panel; control 0 days vs Rab19 0 days and Figure 5—figure supplement 1 Rab3 0 days; most of the Rab mutants after 2 days light such as Rab3 KO and Rab40 KO). These observations should be addressed and discussed in their Results section.

d) In Figure 4, the authors stain for Atg8 and Rab11 to assay for changes in autophagosomes and recycling endosomes, respectively. In RabX1, the authors conclude that there is an increase in Atg8 labeling after exposing adults for 2 days in constant light. The representative figure chosen to represent this increase appears to suggest the opposite. Instead, their appears to be an increase in Atg8 labeling at day 0 but after 2 days of constant light small Atg8 puncta disappear and bigger but lighter blobs appear. To resolve this, the authors should either choose a better representative image or reconsider their interpretation of this data.

---

## [Author Response]

Essential revisions.1) The detailed characterization of RUSH system reveals several concerns and caveats with respect to its use that make aspect of the work is too preliminary and untested to be published as a Resource in eLife. This entire section should therefore be removed.

We have followed this recommendation and removed the original Figure 5, Supplementary figures 4-6 and the associated Results section from the manuscript. To compensate for this section, we have performed and further extended additional experiments beyond the recommended revisions to improve the core of the manuscript: the complete null mutant resource and analyses (new Figure 4, new Figure 7P-S; new Figure 2—figure supplement 1 and 2, Figure 4—figure supplement, Figure 6—figure supplement 1 and Figure 7—figure supplement 1). We feel these revisions result in a more coherent and comprehensive Resource focused entirely on the *rab* mutants and their functional analyses.

2) The sections describing the toolkit of molecularly defined null mutants for all 26 rab genes in *Drosophila* and their characterization are clear and valuable, but also require several additional clarifications and controls before publication.a) The authors test the effects of different temperatures on the development of *Drosophila* in mutants of Rabs enriched in the nervous system. Because mutations are generated via different methods, the genetic backgrounds of those flies should be equalized across all the lines studied (ideally via backcrossing or at least transheterozygotes of independently derived alleles) to exclude unknown variables. The Materials and methods and Results sections do not make it clear if such backcrossing was performed (though the ERG sections indicates that all recordings were performed everything in a w- background). This should be clarified in the manuscript.

This is an important concern that we endeavored to comprehensively address in this revision. All newly generated null mutants in our study were indeed in one of two different genetic background (homologous recombination and CRISPR) that were crossed into the same white minus background, but not further tested in other backgrounds. To test the dependency of the phenotypes on genetic backgrounds, we have performed both types of experiments suggested by the reviewers: “backcrossing or at least transheterozygotes over independently derived alleles” to validate developmental phenotypes. Specifically, we performed both backcrossing over three generations to the same wild type (yw) control and, in addition, we tested all mutant phenotypes in null mutants over deficiencies uncovering the respective *rabs*. As a result, all originally reported developmental phenotypes have now been tested in a total of three genetic backgrounds. Successful validation results from backcrossed homozygous mutants and transheterozygotes over deficiency are presented in full in the new Figure 2—figure supplement 1. The number of successful validations is also shown in the main Figure 2 next to the asterisks that denote significance of the original homozygous mutant: a “2” indicates validation in both other genetic backgrounds (backcrossed and over deficiency), a “1” indicates validation in one of the two other backgrounds, a “0” indicates failure to validate. The results were very encouraging and are described in detail in the Results section for Figure 2: all strongly significant developmental phenotypes of *rab19*, *rab32*, *rabX1* and *rabX4* were validated in both other backgrounds. Most of the more than 20 other significant differences originally observed were also significant in either the backcrossed or transhets over deficiency, thereby showing both some robustness and sensitivity to genetic backgrounds. Only three of the originally (not strongly) significant differences were not validated and are marked with a “0” in Figure 2. In sum, we could validate all main developmental phenotypes, while additionally providing valuable data on the sensitivity of these phenotypes to different genetic backgrounds.

b) The authors measure ERG "on" transients to determine if Rab mutants disrupt synaptic transmission. This is a very important experiment, but the dataset for 2 days light appears highly variable. This is an issue, because several Rabs with "not significant" differences seem to have much higher variances, and therefore there may actually be something going on. This high degree of variability is not observed in controls at 2 days light, or in the 0 days or 4 days dark datasets. Could high levels of variability be a result of neuronal death in Rab mutants exposed to 2 days of light? The authors should explore this issue as a source of variability in their dataset by performing something like a TUNEL assay or EM in these Rab mutants, or at least discuss it.

We fully agree with the importance of the question about the origin of the variability after 2-day light stimulation. Our initial interpretation was that many *rab* mutants represent sensitized backgrounds that can be predicted to exhibit more variability during the most sensitive period (highest dynamic range for differences), given that 2-day light stimulation was determined and chosen to be the most sensitive period to measure differences. To answer the question whether variability based on cell death causes the variability in ERG measurements, we performed immunolabeling with the apoptotic marker Dcp-1 in all mutants analyzed for ERGs (all homozygous viable mutants). We validated Dcp-1 labeling as an assay in the *rdgC^306^* mutant that is known to cause photoreceptor degeneration, and then performed the same immunolabeling of eye sections for the *rab* mutants both before (0 day) and after (2 days) light stimulation. We detected no apoptosis in any of the mutants, indicating that the variability is not due to variability in cell death, but due to variability of functional properties. The data are presented in the new main Figure 4 and the new Figure 4—figure supplement 1 .

As part of the Dcp-1 assay, we used co-labeling of the rhabdomeres, i.e. the densely stacked membranes that house the phototransduction machinery and that are known to be subject to intense membrane trafficking following photoreceptor stimulation. We found that in control after 2 days light stimulation the rhabdomere shape was preserved, but the rhabdomere area increased on average around 30%. Interestingly, all viable *rab* mutants had rhabdomeres indistinguishable from control at 0 days (prior to stimulation), but many exhibited rhabdomere defects after 2 days of light stimulation – all with increased variability. We found this to be useful data both in the context of our comparative functional analyses and as a second functional, stimulus-dependent readout revealing functional variability selectively in the sensitive period of 2 days stimulation. We have therefore included all new data from this experiment as a new main Figure 4 and a new Figure 4—figure supplement 1 .

c) Chaoptin staining is used to assess structural differences in the photoreceptor projections in Rab mutants. The representative images used, which are described in the text as having "no phenotype," appear to have decreased Chaoptin staining (e.g. Figure 4A R1-R6 middle panel; control 0 days vs Rab19 0 days and Figure S3 Rab3 0 days; most of the Rab mutants after 2 days light such as Rab3 KO and Rab40 KO). These observations should be addressed and discussed in their Results section.

We checked all original datasets and can confirm that these apparent differences in the Figure panels are the result of immunolabeling and cross-section visualization based on 3D datasets. These not reproducible phenotypes and the same variability is observed in wild type stainings. We have gone back to the original data and also obtained more imaging data to show better representative pictures in the Figure 5 and the Figure 5—figure supplement 1 .

d) In Figure 4, the authors stain for Atg8 and Rab11 to assay for changes in autophagosomes and recycling endosomes, respectively. In RabX1, the authors conclude that there is an increase in Atg8 labeling after exposing adults for 2 days in constant light. The representative figure chosen to represent this increase appears to suggest the opposite. Instead, their appears to be an increase in Atg8 labeling at day 0 but after 2 days of constant light small Atg8 puncta disappear and bigger but lighter blobs appear. To resolve this, the authors should either choose a better representative image or reconsider their interpretation of this data.

Yes, the *rabX1* figure panel was chosen to highlight the “bigger blobs” pointed at by arrowheads, a phenotype only shown in this mutant. However, going back to the original data, *rabX1* also shows more widespread Atg8 increases, as reported in the text and not well represented in our originally chosen panel. In addition, in the lamina cortex, there really seems to be a redistribution of more smaller Atg8 positive compartments at 0 day into clusters after 2 days of stimulation, as spotted by the reviewer (and not by us in the first analysis!). We do not know what it means, but there is clearly a stimulus-dependent Atg8 (and thereby autophagy) increase and redistribution in the *rabX1* mutant. We have chosen the best representative picture in the revised main Figure and added the description of Atg8 clustering pointed out by the reviewer (now Figure 5B).